# The antimicrobial fibupeptide lugdunin forms water-filled channel structures in lipid membranes

Dominik Ruppelt[1], Marius F. W. Trollmann [2], Taulant Dema [3], Sebastian N. Wirtz[3], Hendrik Flegel[1], Sophia Mönnikes[1], Stephanie Grond [3], Rainer A. Böckmann [2] ✉ & Claudia Steinem [1,4] ✉

Recently, a novel cyclo-heptapeptide composed of alternating D,L-amino acids and a unique thiazolidine heterocycle, called lugdunin, was discovered, which is produced by the nasal and skin commensal *Staphylococcus lugdunensis*. Lugdunin displays potent antimicrobial activity against a broad spectrum of Gram-positive bacteria, including challenging-to-treat methicillin-resistant *Staphylococcus aureus* (MRSA). Lugdunin specifically inhibits target bacteria by dissipating their membrane potential. However, the precise mode of action of this new class of fibupeptides remains largely elusive. Here, we disclose the mechanism by which lugdunin rapidly destabilizes the bacterial membrane potential using an in vitro approach. The peptide strongly partitions into lipid compositions resembling Gram-positive bacterial membranes but less in those harboring the eukaryotic membrane component cholesterol. Upon insertion, lugdunin forms hydrogen-bonded antiparallel β-sheets by the formation of peptide nanotubes, as demonstrated by ATR-FTIR spectroscopy and molecular dynamics simulations. These hydrophilic nanotubes filled with a water wire facilitate not only the translocation of protons but also of monovalent cations as demonstrated by voltage-clamp experiments on black lipid membranes. Collectively, our results provide evidence that the natural fibupeptide lugdunin acts as a peptidic channel that is spontaneously formed by an intricate stacking mechanism, leading to the dissipation of a bacterial cell's membrane potential.

Dissemination of infectious diseases caused by antibiotic-resistant bacterial strains represents a major threat to the global health system[1]. Modern research recently started to aim at identifying new antimicrobial entities originating from natural sources[2,3]. From these sources, the human body microbiome is of special interest as its sensitive interplay between benign and beneficial bacteria is crucial for

human health - more than previously appreciated. Our increasing chemical understanding of interactions within the microbiota allows for translational new concepts for antimicrobial precision therapies and might overcome the limitations of conventional strategies[4]. In particular, bacteria harbored in nutrient-poor environments were found to be a rich source of antimicrobial peptides (AMPs), inhibiting,

[1]Institute of Organic and Biomolecular Chemistry, Georg-August-Universität, Tammannstraße 2, 37077 Göttingen, Germany. [2]Computational Biology, Department Biologie & Erlangen National High Perfomance Computing Center (NHR@FAU), Friedrich-Alexander-Universität Erlangen-Nürnberg, Staudtstraße 5, 91058 Erlangen, Germany. [3]Institute of Organic Chemistry, Eberhard Karls Universität Tübingen, Auf der Morgenstelle 18, 72076 Tübingen, Germany. [4]Max Planck Institute for Dynamics and Self-Organization, Am Faßberg 17, 37077 Göttingen, Germany. ✉e-mail: rainer.boeckmann@fau.de; csteine@gwdg.de

or even killing other bacteria and securing their survival[5–7]. Today, AMPs have emerged to be one of the most promising approaches for the development of potential new antimicrobial entities[8,9]. The novel cyclic AMP lugdunin (structure **1**, Fig. 1) recently isolated from a strain of nasal *Staphylococcus lugdunensis* was identified, representing the first member of the class of fibupeptides which are characterized by a unique thiazolidine heterocycle in the cyclopeptide backbone[10]. In vivo, lugdunin prevents the growth of *Staphylococcus aureus* in the human nose and has shown promising antimicrobial activity against a plethora of other Gram-positive bacteria[10]. In a study with human keratinocytes, lugdunin also displayed synergistic effects with host-derived AMPs and strengthened the immune response by recruiting monocytes and neutrophiles[11]. With the development of a synthetic route towards lugdunin, a comprehensive structure-activity relationship study with a manifold of lugdunin derivatives revealed the essential structural motifs for its antimicrobial activity[12]: The thiazolidine ring is pivotal for the activity and, similarly, L-tryptophan, and D-leucine must not be replaced by alanine. Whereas the alternating D,L-configuration is essential for the antimicrobial activity, the enantiomeric compound is as potent as lugdunin suggesting that for the first attack on bacterial cells, a specific AMP-protein interaction is not required. Instead, lugdunin has been shown to induce dissipation of the bacterial membrane potential of *S. aureus*[12]. As this process is not mediated by the formation of large pores, as observed for other AMPs[13], we proposed that the peptide induces ion transport across the membrane, which was supported by an in vitro proton translocation assay[12]. These results suggest that lugdunin serves as an ion transporter in the membrane, either operating as a carrier or a channel. However, which structure is required to perform this task and the detailed molecular mode of action remain fully elusive.

In this work, we address the question of how lugdunin first attacks bacterial membranes without the need for additional protein components and pursue a combined experimental and molecular dynamics simulations approach using well-defined artificial membrane systems. To unravel whether the specificity of lugdunin for bacterial membranes lies in part in the lipid composition, we first quantify the partitioning of lugdunin into the membrane phase dependent on the lipid composition. ATR-FTIR spectroscopic analysis of the structure of lugdunin and its methylated derivatives (structures **3–5**, Fig. 1) in conjunction with molecular dynamics simulations show that peptide nanotubes are formed in lipid bilayers which are responsible for the transport of protons and other monovalent cations. These results

enabled us to draw a detailed picture of the mode of action of lugdunin in lipid membranes.

## Results and discussion

### Partitioning of lugdunin in POPC membranes

As lugdunin exerts its mechanism of action by targeting the cell membrane[12], peptide partitioning is an utmost important parameter governing its antimicrobial efficiency. To quantify the partitioning of lugdunin between the water and the membrane phase and its position within lipid bilayers, we exploited the intrinsic fluorescent properties of its L-tryptophan residue in the absence and presence of large uni-lamellar vesicles (LUVs). We chose to start with the lipid 1-palmitoyl-2-oleoyl-*sn*-glycero-3-phosphocholine (POPC) to relate our previous initial results on the proton translocation capability of lugdunin in POPC LUVs[12] with its partitioning into these bilayers. Such simple model membranes enable one to investigate peptide-membrane interactions in a well-defined system without the influence of, e.g., bacterial proteins[14].

In an aqueous buffer, the fluorescence emission maximum of lugdunin was located at $(354 \pm 1)$ nm indicative of a polar environment of the tryptophan residue (Fig. 2a, black curve)[15]. Upon increasing the lipid concentration, the intensity at the emission maximum increased and the position of the maximum was blue-shifted to lower wavelengths. A maximum blue shift of $(18 \pm 2)$ nm in the presence of 2 mM POPC was determined (Fig. 2a, blue curve, Fig. 2b), accompanied by an increase in the maximum fluorescence intensity (Fig. 2a, c). The observed blue shift indicates the partitioning of the L-tryptophan's indole side chain into the hydrophobic part of the vesicular lipid membrane[16–19]. The maximum value of the blue shift depends, among others, on the insertion depth of the L-tryptophan residue into the hydrophobic core of the membrane and is used to estimate the position of the peptide. With a blue shift of $(18 \pm 2)$ nm, the L-tryptophan residue of lugdunin is located in between the hydrophobic core and the lipid headgroup region. For comparison, the blue shift found for the cationic peptide melittin was around 20 nm characteristic for penetration deep into the hydrophobic membrane core, whereas for the antimicrobial peptide cecropin A a blue shift of 15 nm was determined and the peptide was located near the lipid head group region[20,21].

The change in fluorescence emission further enables one to quantify the partitioning of lugdunin between the aqueous and the membrane phase[22]. Fitting Eq. (3) to the data yielded a partition coefficient of $K_{\chi,\mathrm{exp}} = (9 \pm 2) \cdot 10^5$ $(\log_{10}(K_\chi)_{\mathrm{exp}} = 6.0 \pm 0.1)$ for POPC, demonstrating that lugdunin preferentially partitions into the membrane phase.

To further corroborate these findings of a strong membrane partitioning of lugdunin, collisional quenching experiments with water-soluble iodide ions were performed reporting on the water-accessibility of the L-tryptophan residue[23]. From the Stern−Volmer plots (Fig. 2d), it is evident that quenching is less pronounced in the presence of POPC vesicles, supporting the hypothesis that the tryptophan residue is buried inside the lipid bilayer. The corresponding Stern−Volmer constant reads $(7.9 \pm 0.2)$ M$^{-1}$ in buffer solution and was reduced by roughly one-half to $(4.5 \pm 0.2)$ M$^{-1}$ in the presence of POPC vesicles.

To relate our experimental findings to the molecular structure of lugdunin, we further investigated the spontaneous insertion of lugdunin into lipid membranes by employing all-atom molecular dynamics (MD) simulations. Peptide monomers were initially randomly placed in close proximity to the surface of POPC bilayers (peptide-to-lipid ratio 1:40 (*n/n*), four replicas each). All peptides inserted into the POPC membrane very fast (1–2 µs, Supplementary Fig. 17a, top row). The initial attachment and anchoring of lugdunin to the membrane interface were driven by L-tryptophan, followed by penetration of the hydrophobic residues and of the thiazolidine

| | | |
|---|---|---|
| **1** | Lugdunin | R₁ = D-Val; R₂, R₃, R₄ = H |
| **2** | 3,6-Ditrp-lugdunin | R₁ = D-Trp; R₂, R₃, R₄ = H |
| **3** | 3-Methyl-3,6-ditrp-lugdunin | R₁ = D-Trp; R₂ = CH₃; R₃, R₄ = H |
| **4** | 4-Methyl-3,6-ditrp-lugdunin | R₁ = D-Trp; R₂, R₄ = H; R₃ = CH₃ |
| **5** | 6-Methyl-3,6-ditrp-lugdunin | R₁ = D-Trp; R₂, R₃ = H; R₄ = CH₃ |

**Fig. 1 | Chemical structure of lugdunin and lugdunin analogs.** R¹ only refers to the respective amino acid side chain.

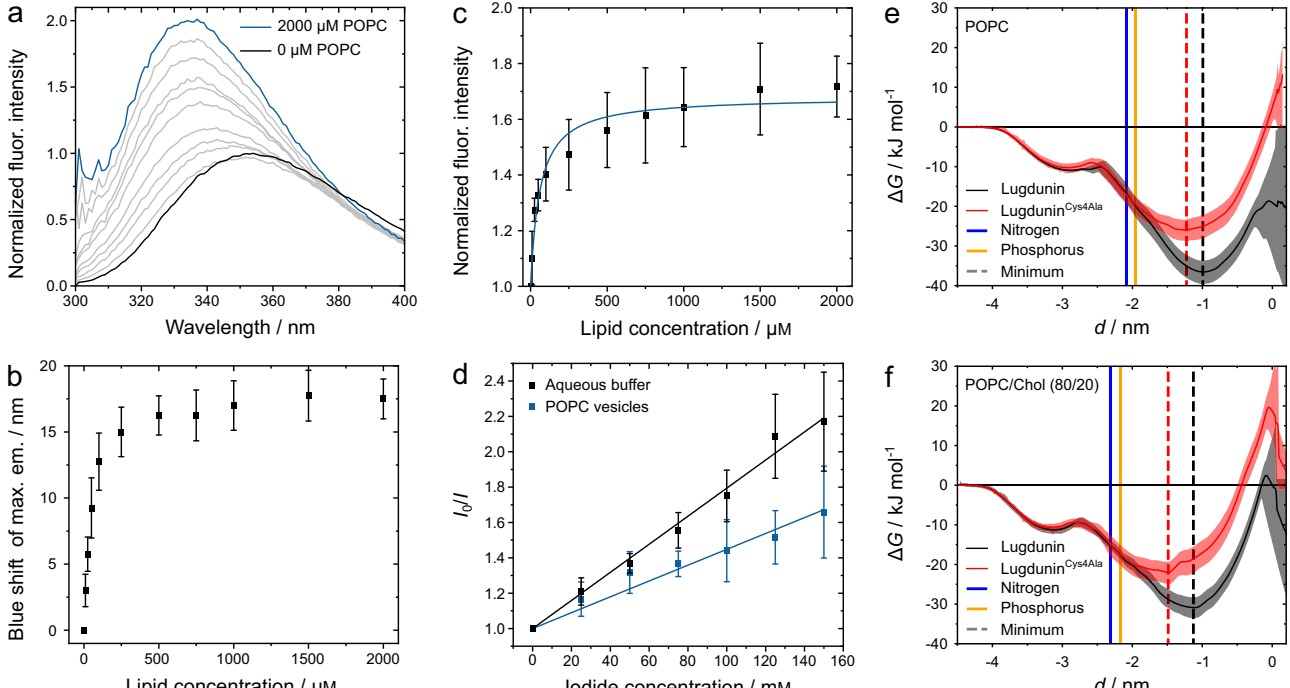

**Fig. 2 | Insertion of lugdunin into the membrane phase. a** Fluorescence emission spectra of lugdunin (10 μM, black curve) obtained at different lipid concentrations (gray curves) up to a concentration of 2 mM POPC (blue curve). **b** Mean blue shift of the maximum emission wavelength dependent on the lipid concentration ($n = 4$ independent experiments). The error bars are the standard deviation of the mean. **c** Increase in fluorescence intensity upon partitioning of lugdunin into lipid membranes ($n = 4$ independent experiments, $R^2 = 0.96$). The error bars are the standard deviation of the mean. **d** Mean Stern–Volmer plots of the tryptophan quenching of

lugdunin (5 μM) with iodide in the absence ($R^2 = 0.99$) and presence of POPC vesicles ($R^2 = 0.99$, 500 μM, peptide-to-lipid ratio 1:100, $n = 6$ independent experiments). The error bars are the standard deviation of the mean. **e, f** Potential of mean force profiles for the insertion of lugdunin (**1**) and the analog lugdunin$^{Cys4Ala}$ (Supplementary Fig. 19a) into POPC (**e**) and POPC/Chol (80:20, $n/n$) model membranes (**f**). Shaded areas represent 95% confidence intervals derived from bootstrapping of umbrella simulation histograms.

moiety into the hydrophobic core of the membrane (Supplementary Fig. 17b). The observation that L-tryptophan inserts first might explain why this amino acid must not be replaced by, e.g., alanine[12].

## Lugdunin partitioning as a function of the lipid composition

Lugdunin displays a strong antimicrobial activity against Gram-positive bacterial cells, but not against a variety of human-derived cells like erythrocytes, neutrophils, and keratinocytes[10,11]. To become antimicrobially active, the first step requires the partitioning of the peptide into a lipid bilayer. The specificity of lugdunin's antimicrobial activity thus raises the question of whether the lipid composition influences the peptide partitioning into a membrane. To address the main components of Gram-positive bacteria, we added 1-palmitoyl-2-oleoyl-*sn*-glycero-3-phospho(1'-*rac*-glycerol) (POPG) or 1',3'-bis[1,2-dioleoyl-*sn*-glycero-3-phospho]-glycerol (cardiolipin, CL). As an essential component of eukaryotic cell membranes, we added cholesterol (Chol)[24,25]. The resulting maximum blue shifts and partition coefficients for the different lipids are summarized in Table 1 (Supplementary Fig. 18).

The results demonstrate that the partitioning of lugdunin into lipid membranes is unaffected by the negative charge introduced by POPG and only slightly diminished in the presence of cardiolipin. A similarly strong partitioning and a pronounced blue shift were also observed in lipid membranes consisting only of POPG and cardiolipin, mimicking the cell membrane of *S. aureus*. Also, the partition coefficient for a Gram-positive model membrane determined from all-atom MD simulations was comparably large with $\log_{10}(K_\chi)_{sim} = 5.1 \pm 0.6$ (Table 1). These findings are in line with the reported strong antibacterial activity of lugdunin[10].

However, cholesterol affects peptide partitioning. In the case of 20 mol% cholesterol in POPC, the partition coefficient was reduced by one order of magnitude as compared to pure POPC to $\log_{10}(K_\chi)_{exp} = 5.0 \pm 0.2$ indicating a reduced affinity for eukaryotic lipid membranes. Strikingly, the determined fluorescent blue shift was with $(12 \pm 2)$ nm less pronounced compared to vesicles lacking cholesterol. This suggests that lugdunin is not able to penetrate deeply into the membrane core but is rather located in the vicinity of the lipid head-group region. The reduced partitioning and the localization at the membrane interface were also reflected in the increased Stern–Volmer constant $((5.7 \pm 0.4)\,\text{M}^{-1})$ showing the increased water accessibility of the L-tryptophan residue.

The observation that cholesterol impedes the insertion of lugdunin is substantiated by comprehensive molecular dynamics simulations: Notably, in unbiased simulations, the peptide's insertion was significantly diminished in membranes containing 30 mol% cholesterol (Supplementary Fig. 17a, bottom row). This phenomenon was further quantified through an evaluation of the potential of mean force (PMF) associated with lugdunin's membrane insertion based on data from umbrella sampling simulations. The PMF profiles for lugdunin partitioning into POPC and POPC/Chol (80:20) membranes are depicted in Fig. 2e, f. The PMF displays a shallow minimum of $\Delta G \approx -11\,\text{kJ}\,\text{mol}^{-1}$ at the membrane surface, followed by a deeper, global minimum of $\Delta G \approx -36.6\,\text{kJ}\,\text{mol}^{-1}$ for pure POPC membranes which is increased to $\Delta G \approx -26.0\,\text{kJ}\,\text{mol}^{-1}$ in presence of 20 mol% cholesterol. This minimum is located approximately 1 nm beneath the lipid phosphate groups. The associated partition coefficient was $\log_{10}(K_\chi)_{sim} = 5.5 \pm 0.5$ for pure POPC and exhibits an eight-fold reduction to $\log_{10}(K_\chi)_{sim} = 4.6 \pm 0.4$ in the cholesterol-enriched environment, in very good agreement to

**Table 1 | Partitioning of lugdunin into lipid membranes**

| Medium | $\Delta\lambda_{max}$/nm | $\log_{10}(K_\chi)_{exp}$ | $\log_{10}(K_\chi)_{sim}$ | $K_{SV}$ / $\mathrm{M}^{-1}$ |
|---|---|---|---|---|
| Buffer | – | – | – | $7.9 \pm 0.2$ |
| POPC | $18 \pm 2$ | $6.0 \pm 0.1$ | $5.5 \pm 0.5$ | $4.5 \pm 0.2$ |
| POPC/POPG (75:25) | $19 \pm 3$ | $5.9 \pm 0.1$ | – | – |
| POPC/POPG (50:50) | $17 \pm 3$ | $6.0 \pm 0.1$ | – | – |
| POPC/CL (50:50) | $17 \pm 3$ | $5.8 \pm 0.1$ | – | – |
| POPG/CL (50:50) | $19 \pm 3$ | $5.7 \pm 0.2$ | $5.1 \pm 0.6^a$ | – |
| POPC/Chol (90:10) | $13 \pm 2$ | $5.2 \pm 0.1$ | – | $4.9 \pm 0.3$ |
| POPC/Chol (80:20) | $12 \pm 2$ | $5.0 \pm 0.2$ | $4.6 \pm 0.4$ | $5.7 \pm 0.4$ |

Blue shift $\Delta\lambda_{max}$, partition coefficients $K_\chi$ (determined from experiment (exp) and MD simulation (sim)) and Stern–Volmer constants $K_{SV}$ for lugdunin dependent on the lipid composition ($n \geq 2$ independent experiments, mean ± standard deviation for experiments, mean ± 95% confidence interval for simulations).
$^a$Gram+ composition (see Supplementary Fig. 16).

experiment (Table 1). Furthermore, the presence of cholesterol produces a more pronounced energy barrier inside the membrane, hampering lugdunin's efficient partitioning into the hydrophobic membrane core region.

As the thiazolidine ring of lugdunin is a unique structural element, which might be pivotal for the peptide insertion, we also investigated a lugdunin-like derivative in which the thiazolidine moiety was replaced by an alanine (Lugdunin[Cys4Ala], Supplementary Fig. 19). Noteworthy, this structural change resulted in a significant reduction of the partition coefficient by almost two orders of magnitude to $\log_{10}(K_\chi)_{sim} = 3.8 \pm 0.5$ (Supplementary Fig. 20). Concurrently, the energy minimum for this analog is displaced outward by 2.4 Å for POPC and 3.6 Å for POPC/Chol (80:20). This pronounced shift emphasizes that the thiazolidine moiety may play a critical role in the interaction dynamics between lugdunin and lipid membranes.

Cholesterol is known to increase the rigidity of fluid fully saturated or mono-unsaturated membranes affecting its structural assembly by a lipid condensing effect[26–29]. This is expected to interfere with peptide insertion, which is in line with the previously reported impeding influence of cholesterol on the insertion of various AMPs[30]. For example, the activity of gramicidin S and S-thanatin was strongly reduced in the presence of cholesterol-containing vesicles[31,32]. Our results are in line with these findings and demonstrate that the cholesterol-driven lipid condensing and the related increased order of the acyl chains impede lugdunin insertion into the membrane possibly inhibiting exertion of its mode of action from inside the lipid bilayer. However, the reduced partitioning, which is still significantly large, is not sufficient to understand lugdunin's structure within the membrane and its antibacterial activity.

### Structural analysis of lugdunin in lipid membranes

Once in the membrane, the question needs to be addressed which structure lugdunin forms that allows proton translocation across the membrane. To gather structural information, we employed ATR-FTIR spectroscopy. Ordered multi-bilayers composed of 1,2-dimyristoyl-*sn*-glycero-3-phosphocholine (DMPC) and lugdunin were produced on an ATR-IR crystal. DMPC was chosen instead of POPC as it is known to form highly ordered lipid bilayers (Supplementary Table 8) which were already previously used to investigate membrane-embedded proteins and peptides[33,34]. Besides the absorption bands characteristic for lipids (e.g., $\nu_{as}(N(CH_3)_3^+)$ at 3027 cm$^{-1}$, $\nu_{as}(CH_3)$ at 2957 cm$^{-1}$, $\nu_{as}(CH_2)$ at 2919 cm$^{-1}$, $\nu_s(CH_2)$ at 2850 cm$^{-1}$, $\nu(C=O)$ at 1738 cm$^{-1}$)[35], the IR-spectrum (Fig. 3a) also displays the characteristic absorption bands of the peptide bonds i.e, the amide A (3300–3500 cm$^{-1}$), amide I (1600–1700 cm$^{-1}$) and amide II band (1470–1570 cm$^{-1}$). The position of the amide I band reports on the secondary structure of the peptide. We extracted the exact peak positions of the amide I band (Fig. 3b) from second derivative spectra revealing peaks at 1641 cm$^{-1}$ (perpendicular component, strong absorption) and 1683 cm$^{-1}$ (parallel

component, weak absorption) characteristic for a tightly hydrogen-bonded antiparallel β-sheet structure[36].

The amide II region also shows a characteristic β-sheet peak at 1543 cm$^{-1}$ (Fig. 3a)[37]. The amide A peak located at 3282 cm$^{-1}$ supports a tight hydrogen bonding of the N-H groups of the peptide backbone (Fig. 3a). Assuming intermolecular hydrogen bond interactions between individual peptide molecules, the average N−O distance in a hydrogen bond is estimated by using Krimm's correlation between the position of the amide A band and the hydrogen bond length[38]. Following this, an amide A peak at 3282 cm$^{-1}$ translates to an N−O distance of around 2.88 Å. To extract the fraction of the secondary structural elements from the amide I band, the peak integrals were taken after deconvoluting the spectra (Table 2). For peptide-to-lipid ratios of 1:40 and 1:30 the deconvolution of the amide I band results in 100% antiparallel β-sheet structure. At peptide-to-lipid ratios larger than 1:20 an additional peak in the amide I region emerges at 1624 cm$^{-1}$ (Supplementary Fig. 21, Table 2). We conclude that at this high peptide concentrations, lugdunin aggregates are formed between the multi-bilayer stacks which do not reflect the structure within the lipid bilayers.

The tightly hydrogen-bonded antiparallel β-sheet structure of lugdunin observed by ATR-IR spectroscopy reminded us on the reported structures of synthetic cyclopeptides with D,L-configuration[34,39–42]. The IR spectrum of, e.g., *cyclo*[(L-Trp-D-Leu)$_3$-L-Gln-D-Leu] in DMPC multi-bilayers displayed amide I peaks at 1635 cm$^{-1}$ and 1688 cm$^{-1}$, an amide II peak at 1538 cm$^{-1}$, and an amide A peak at 3281 cm$^{-1}$ [43]. These characteristic IR bands are very similar to those found for lugdunin and are indicative of antiparallel β-sheet structures. There are, however, significant structural differences between the synthetic D,L-cyclopeptides and lugdunin. Lugdunin, as a member of the novel class of fibupeptides, is composed of an odd number of amino acids and not an even number like the synthetic cyclopeptides and harbors a unique thiazolidine heterocycle. The even number of amino acids of the synthetic cyclopeptides favors a planar ring geometry resulting in an assembly of ring stacks that is connected via strong and defined hydrogen bonds[44]. These so-called nanotubes have a more polar inner pore and apolar amino acid side chains facing the membrane. Embedded in a membrane, they allowed the passage of small molecules and ions[45] and they could act as a channel with the potential of serving as an ionophore-like AMP[46].

In the case of lugdunin, the thiazolidine moiety might disturb the planarity of the ring. However, the characteristic IR-bands still suggest that a similar structure composed of ring stacks connected via hydrogen bonds is formed.

### Orientation of lugdunin in lipid membranes

To gather further support for a ring stack, forming a nanotube structure, we determined the orientation of the peptide rings in the DPMC multi-bilayers using polarized ATR-FTIR spectroscopy. We recorded polarized ATR-FTIR spectra of the lipid acyl chains (Fig. 3c) and the

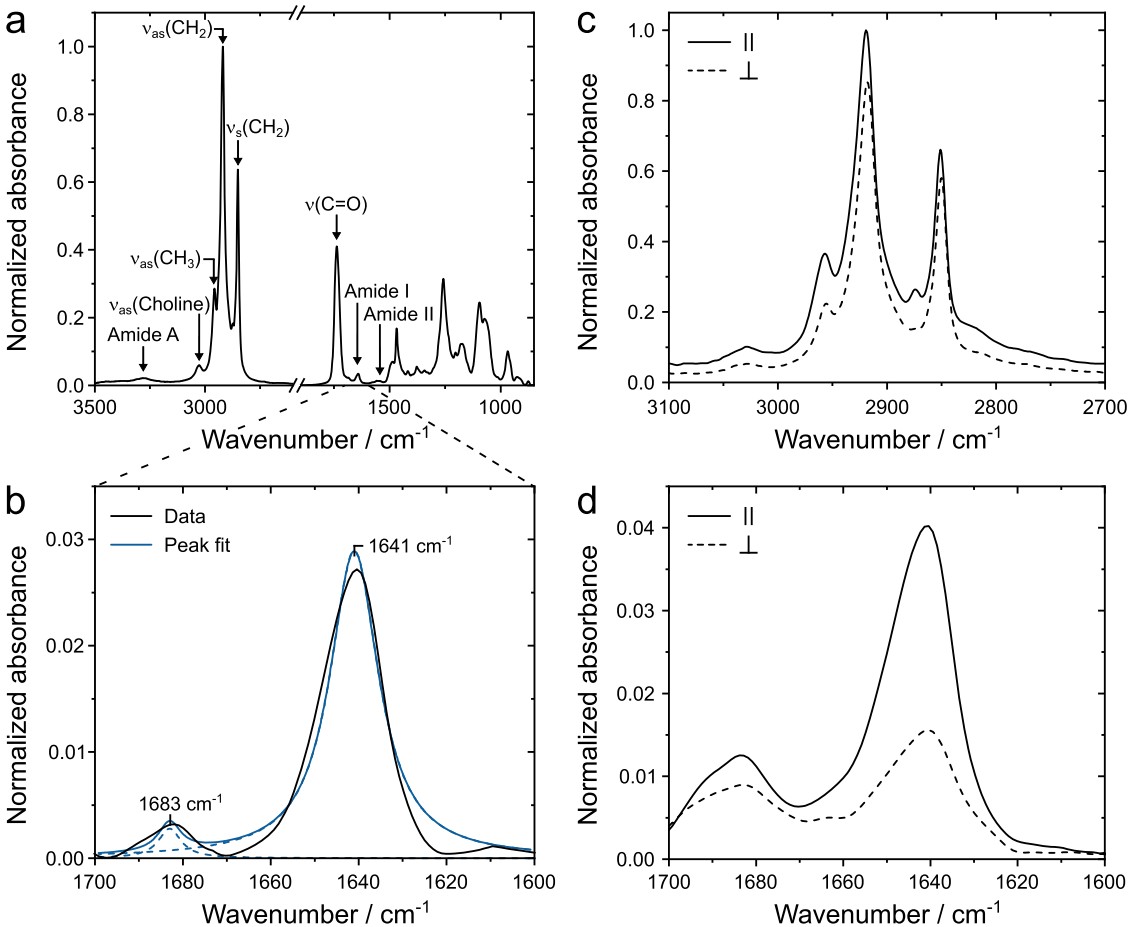

**Fig. 3 | ATR-FTIR spectra of DMPC multi-bilayers with reconstituted lugdunin.**
**a** ATR-FTIR spectrum of DMPC multi-bilayers with lugdunin. **b** Amide I region of the spectrum depicted in **a** (black) with peak deconvolution (blue, $R^2 = 0.97$). A nominal peptide-to-lipid ratio of 1:40 (*n/n*) was used. **c** Polarized methylene stretching modes of a pure DMPC multi-bilayer. **d** Polarized amide I region of lugdunin in DMPC multi-bilayers with a peptide-to-lipid ratio of 1:40 (*n/n*).

amide I region (Fig. 3d) to determine the orientation of the lipid molecules and lugdunin in the multi-bilayers.

The exact experimental details are described in Supplementary Fig. 10 and Supplementary Table 1. The lipid order was assessed by measuring the dichroic ratio of the antisymmetric and symmetric CH$_2$-stretching vibrations, whose transition dipole moments are aligned perpendicular to the axis of the all-*trans* hydrocarbon chain. In the absence of lugdunin, DMPC molecules showed an average tilt angle of $(27.5 \pm 0.6)°$, which is in good agreement with previously reported values obtained for lipid multi-bilayer systems[42]. The addition of lugdunin increased the tilt angle between 1° and 7° depending on the peptide concentration. For lugdunin orientation, we evaluated the dichroic ratio of the amide I peak at ≈1640 cm$^{-1}$ (perpendicular component, Fig. 3d). Assuming a nanotube structure of stacked lugdunin monomers, similar to that described by Kim et al.[43], the perpendicular component of the amide I transition dipole moment would be oriented parallel to the central axis of the nanotube.

The determined dichroic ratio of the amide I band indicates that the central axis of the hydrogen-bonded peptide rings is oriented in parallel to the crystal plate normal. A similar result was found when evaluating the amide A band, which is mainly composed of the N−H-stretching vibration. The tilt angle of the hydrogen-bonded peptide rings was determined to be $(44 \pm 2)°$ relative to the crystal surface normal, which translates into a tilt angle relative to the membrane normal of $(10 \pm 2)°$ (Supplementary Table 9). The tilt angle of around 10° is similar to angles found for the β-helix of gramicidin A in DMPC multi-bilayers (15°) and that of the synthetic D,L-cyclopeptide *cyclo*[(L-Trp-D-Leu)$_3$-L-Gln-D-Leu] (7°)[43,47] further corroborating our hypothesis that lugdunin forms nanotubes in lipid membranes.

**Structure and orientation of lugdunin analogs**
Our results strongly suggest that nanotubes are the main structure formed by lugdunin in model membranes, which may act as channels and might be responsible for the observed in vitro proton

## Table 2 | Secondary structure of lugdunin and analogs

| Peptide | $\tilde{\nu}$ / cm$^{-1}$ | Assignment | Relative area / % |
|---|---|---|---|
| 1[a] | 1641, 1683 | Antiparallel β-sheet | 100 |
| 1 | 1639, 1682 | Antiparallel β-sheet | 49 ± 11 |
| | 1624 | Aggregation | 50 ± 11 |
| 2 | 1641, 1683 | Antiparallel β-sheet | 73 ± 7 |
| | 1655 | Unordered | 17 ± 12 |
| | 1626 | Aggregation | 9 ± 5 |
| 3 | 1640, 1680 | Antiparallel β-sheet | 38 ± 4 |
| | 1655 | Unordered | 62 ± 4 |
| 4 | 1636, 1684 | Antiparallel β-sheet | 34 ± 2 |
| | 1654 | Unordered | 66 ± 2 |
| 5 | 1636, 1677 | Antiparallel β-sheet | 36 ± 5 |
| | 1645, 1654 | Unordered | 64 ± 5 |

Assignment and relative area of the IR bands attributed to the secondary structure of lugdunin and derivatives **2–5** (*n* ≥ 2 independent experiments, mean ± standard deviation). If not indicated otherwise, data are for peptide-to-lipid ratios of 1:10 (*n/n*).
[a]Data are given for a peptide-to-lipid ratio of 1:40 (*n/n*).

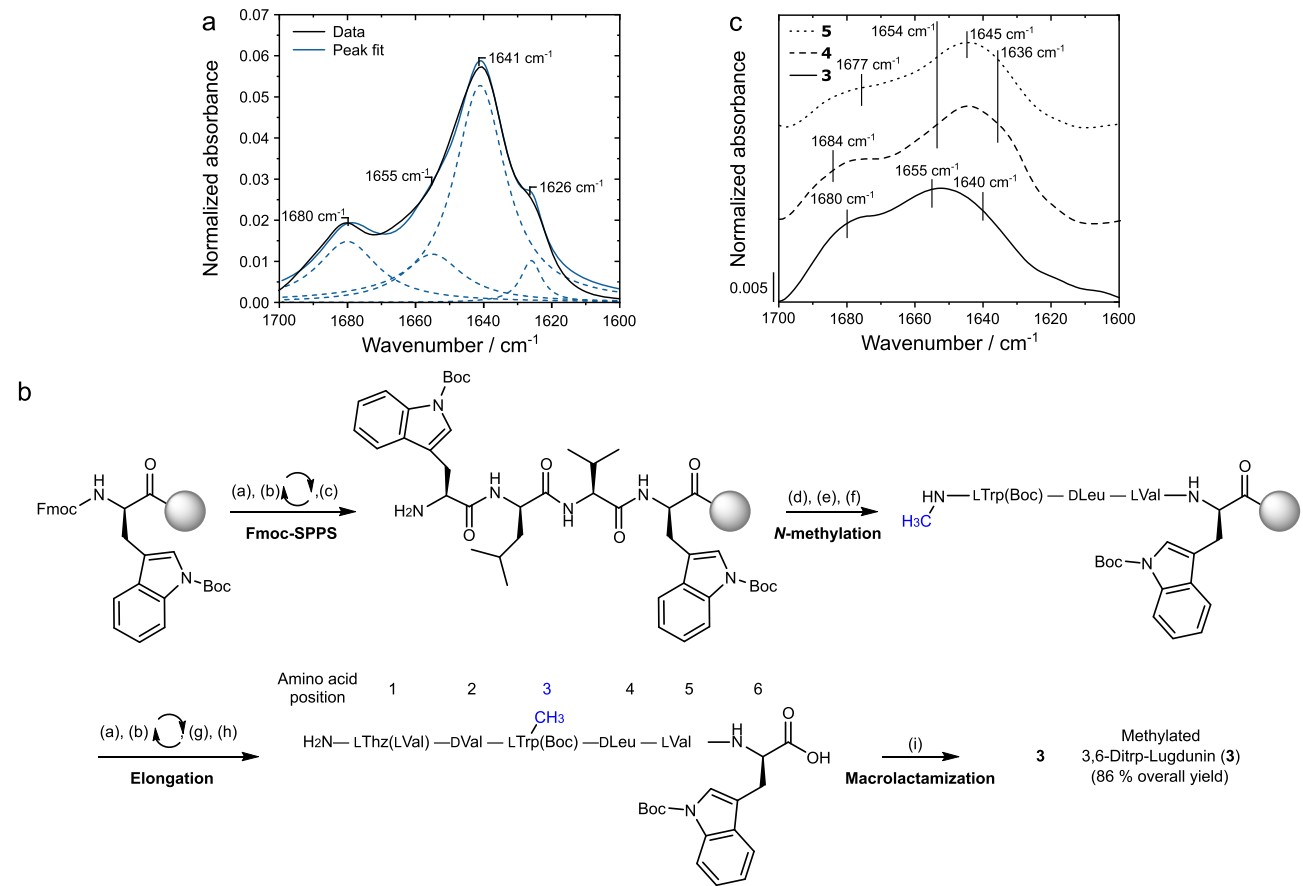

**Fig. 4 | Synthesis and analysis of lugdunin derivatives 3-5 compared to 3,6-ditryptophan-lugdunin. a** ATR-FTIR spectra of DMPC multi-bilayers with 3,6-ditryptophan-lugdunin (black, peptide-to-lipid ratio 1:10 ($n/n$)) and peak deconvolutions (blue, $R^2$ = 0.99). **b** Exemplary solid-phase peptide synthesis of methylated lugdunin analog **3** with the amino acid position in blue. (a) DBU (2%), morpholine (10%) in DMF. (b) Fmoc-ʟ-Val (6 eq.), HATU (6 eq.), HOBt (6 eq.), *N*-methylmorpholine (8 eq.). (c) step (a) then (b) with Fmoc-ᴅ-Leu-OH,

Fmoc-ʟ-Trp(Boc)-OH, respectively. (d) *o*NBSCl (4 eq.), collidine (5 eq.) in DCM. (e) Dimethyl sulfate (3 eq.), MTBD (3 eq.) in DMF. (f) DBU (5 eq.), 2-mercaptoethanol (10 eq.) in DMF, (g) step (a) then (b) with Fmoc-ᴅ-Val-OH, Fmoc-ʟ-Thz(ʟ-Val)-OH, respectively. (h) TFA (90%), TIPS (5%), water (5%). (i) HATU (4 eq.), HOAt (6 eq.), DIPEA (8 eq.). Synthesis of **4** and **5** were performed accordingly. **c** ATR-FTIR spectra of DMPC multi-bilayers with methylated 3,6-ditryptophan-lugdunin analogs **3**, **4**, and **5** (peptide-to-lipid ratio 1:10 ($n/n$)).

translocation activity and in vivo activity. To further support our hypothesis, we analyzed the secondary structure of 3,6-ditryptophan-lugdunin (structure **2**, Fig. 1) by ATR-FTIR spectroscopy, as this derivative showed a two-fold increased antimicrobial activity in in vivo assays compared to lugdunin[12,48]. Indeed, 3,6-ditryptophan-lugdunin also adopted an antiparallel β-sheet structure with peaks located at 1641 cm⁻¹ and 1680 cm⁻¹ (Fig. 4a, Table 2). By polarized ATR-FTIR spectroscopy, the relative tilt angle of the hydrogen-bonded peptide rings of 3,6-ditryptophan-lugdunin was determined to (2 ± 2)° relative to the membrane normal. This finding suggests that the increased bactericidal activity of this analog is related to a well-aligned nanotube structure.

It is known that the planar aromatic structure of the indole ring system of tryptophan favors its localization at the lipid-water-interface of membranes in peptides and proteins, thus stabilizing the interfacial region of proteins[49–51]. Following this, it is conceivable that the addition of a second tryptophan residue promotes the alignment of lugdunin β-sheet nanotubes. The IR peak at 1626 cm⁻¹ that we found for lugdunin at peptide-to-lipid ratios larger than 1:20 and which we attributed to aggregated peptides between the multi-bilayer stacks, is almost negligible for 3,6-ditryptophan-lugdunin demonstrating that this aggregation is not associated with lugdunin's antibacterial activity. Notably, this finding enabled us to use this compound and its analogs at a high peptide-to-lipid ratio of 1:10 leading to a high signal-to-noise ratio for further structural studies.

As the ability of the peptide rings to participate in hydrogen bonds between the cyclic monomers is a prerequisite for nanotube formation, we tested our hypothesis by synthesizing *N*-methylated derivatives of 3,6-ditryptophan-lugdunin (Fig. 4b). Methylation at the peptide backbone is expected to interfere with the ability of the peptide to form extensive hydrogen-bonded β-sheet structures[45,52,53]. 3,6-Ditryptophan-lugdunin was methylated at positions 3, 4 and 6, respectively, using an on-resin *N*-methylation approach[54] (Fig. 1, structures **3–5**, experimental results are given in Supplementary Figs. 1–9) and their IR-spectral properties were investigated. All derivatives displayed a secondary structure consisting of 65% unordered structure (with a peak at around 1655 cm⁻¹) and 35% antiparallel β-sheets indicated by peaks at ≈1638 cm⁻¹ and ≈1682 cm⁻¹ (Fig. 4c, Table 2). Compared to the non-methylated compound, methylation increased the amount of unordered structural elements considerably (>60%). With these findings, it is evident that a change in the hydrogen-bonding capability of lugdunin results in a fundamental change of the observed structure in lipid membranes, further strengthening our idea of a hydrogen-bonded β-sheet.

## Ion translocation activity of lugdunin and analogs

To relate the structural information gathered with the activity of lugdunin, we analyzed the ion translocation capability of lugdunin and the analogs. Proton translocation in the presence of lugdunin across neat POPC bilayers was monitored as a decrease in pyranine fluorescence

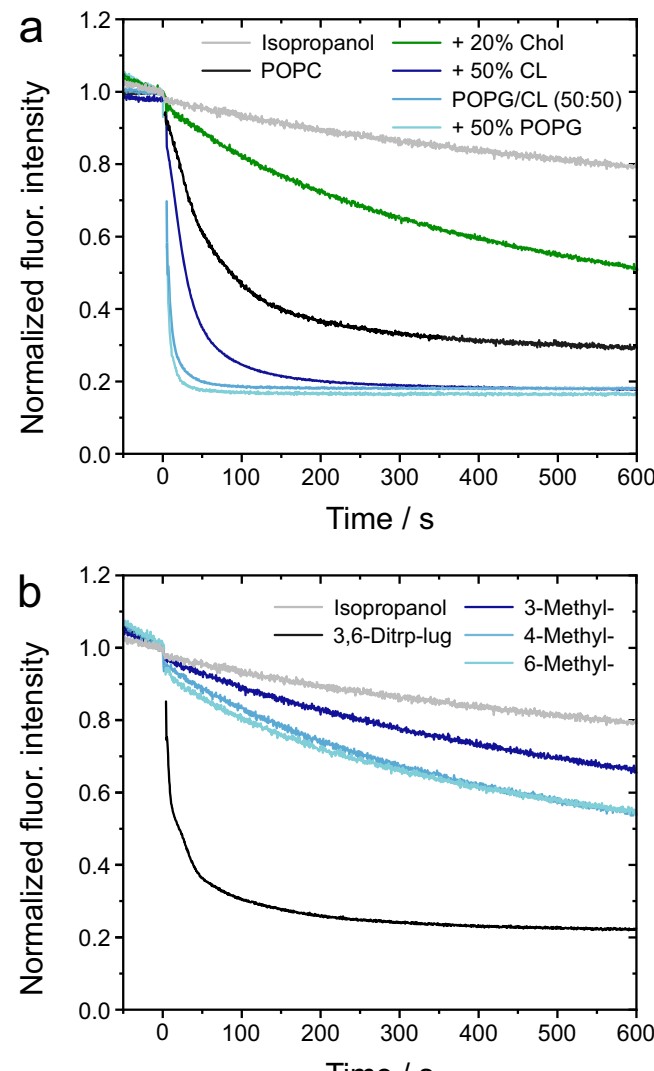

**Fig. 5 | Proton transport activity of lugdunin and its methylated analogs 3–5 measured with pyranine entrapped in lipid vesicles. a** Proton translocation induced by lugdunin as a function of the lipid composition. **b** Proton translocation induced by 3,6-ditryptophan-lugdunin and methylated analogs **3**, **4**, and **5**. A nominal peptide-to-lipid ratio of 1:250 (*n/n*) was applied in all cases.

upon the addition of lugdunin to vesicles exposed to a pH gradient. The observed results are in agreement with our previous findings (Fig. 5a)[12]. However, the observed acidification of the vesicle lumen could also be explained by a chloride ion transport as they accompany the applied pH gradient. To rule out that chloride is transported, we performed vesicle assays leveraging a lucigenin-based approach[55]. We did not find any significant influence of lugdunin on the resulting fluorescence time courses (Supplementary Fig. 22) demonstrating that lugdunin does not transport chloride ions but protons.

To analyze whether the lipid composition alters the proton translocation activity, we employed vesicles composed of different lipid compositions. Adding negatively charged POPG (+50% POPG) or cardiolipin (+50% CL) characteristic for Gram-positive bacteria increased the kinetics of proton transfer significantly (Fig. 5a). The same result was found for bilayers composed only of POPG and cardiolipin (POPG/CL, 50:50) resembling the lipid composition of *S. aureus*. As the partition coefficients of lugdunin are very similar for POPC, POPC/POPG, POPC/cardiolipin and POPG/cardiolipin bilayers (Table 1), the faster kinetics might be attributed to the locally increased

proton concentration at the negatively charged membrane surface[56]. In the case of vesicles composed of POPC/Chol (+20% Chol), the proton transport activity was, however, significantly reduced (Fig. 5a). This reduction agrees with the observed hampered insertion of lugdunin. Cholesterol has multifaceted effects on the membrane's physical properties, inducing a membrane stiffening in saturated or mono-unsaturated membranes. Probably even more important is the increased membrane thickness upon cholesterol addition[26–29]. The increased thickness necessitates additional stacked lugdunin molecules to traverse the hydrophobic core of the membrane. Additionally, there is an elevated free energy barrier associated with lugdunin's partitioning into the membrane core region in the presence of cholesterol (Fig. 2e, f). Collectively, these factors diminish the likelihood of forming peptide nanotubes, thus providing a first rationale for lugdunin's bacterial membrane selectivity.

If peptide nanotubes are the active species in a lipid membrane capable of transporting protons, only those lugdunin derivatives that partition into the bilayer and can form hydrogen bonds are expected to permeabilize the membrane for protons. Thus, we supported the idea of lugdunin nanotubes by employing 3,6-ditryptophan-lugdunin and its methylated peptide analogs **3–5** (Fig. 5b). 3,6-Ditryptophan-lugdunin displayed a slightly increased proton translocation which is in line with its increased bioactivity[12] and our observation from ATR-FTIR experiments suggesting that this derivative forms well-aligned nanotubes in lipid membranes. Even though the methylated peptides exhibit the appropriate hydrophobic surface to partition into bilayers, they showed a significantly reduced proton transport activity across lipid membranes and a considerably reduced bioactivity against *S. aureus* compared to 3,6-ditryptophan-lugdunin (minimal inhibitory concentrations >100 μM). Their significantly reduced proton transport activity suggests that the hydrogen-bonded antiparallel β-sheet structure found for lugdunin and 3,6-ditryptophan-lugdunin, but not for methylated analogs **3**, **4**, and **5**, is pivotal for the activity of the lugdunin fibupeptides. Conclusively, the reduced ability of the methylated analogs **3**, **4**, and **5** to form ordered intermolecular hydrogen bond-connected ring stacks prevents the formation of functional peptide nanotubes resulting in a reduced or even almost abolished proton translocation activity.

## Structure, orientation, and properties of lugdunin nanotubes in silico

To provide a molecular picture of the lugdunin nanotubes in a lipid membrane based on our structural information, we modeled different lugdunin nanotubes built of hydrogen-bonded antiparallel β-sheets with 3–5 lugdunin cyclopeptides, arranged in tryptophan *trans-* and *gauche*-configuration (Supplementary Fig. 23). These nanotubes were inserted into different lipid membranes (DMPC, POPC, POPC/Chol, and 1,2-dioleoyl-*sn*-glycero-3-phosphocholine (DOPC)) and were subjected to microsecond-long MD simulations (with nine replicas each). The nanotube stability was assessed by analyzing the number of intermolecular hydrogen bonds or water bridges (≤0.4 nm acceptor-donor distance, ≤30° between acceptor, donor, and hydrogen atom) between the backbones of neighbored peptides (Supplementary Fig. 24). The most stable configurations were trimers in DMPC (at $T = 310$ K), tetramers in POPC and DOPC (Fig. 6a), and pentamers in POPC/Chol membranes (all at $T = 298$ K). Notably, intermolecular interactions within the membrane were more stable than near the membrane surface. Representative snapshots displaying the hydrogen bond network for a trimeric channel are shown in Supplementary Fig. 25.

The lugdunin nanotubes spanned the hydrophobic core of the membrane and were anchored to the membrane interface domain by their amphipathic tryptophan residues, exhibiting an average tilt of approximately 20–30° with respect to the membrane normal (Supplementary Fig. 26). These nanotubes facilitate the formation of a

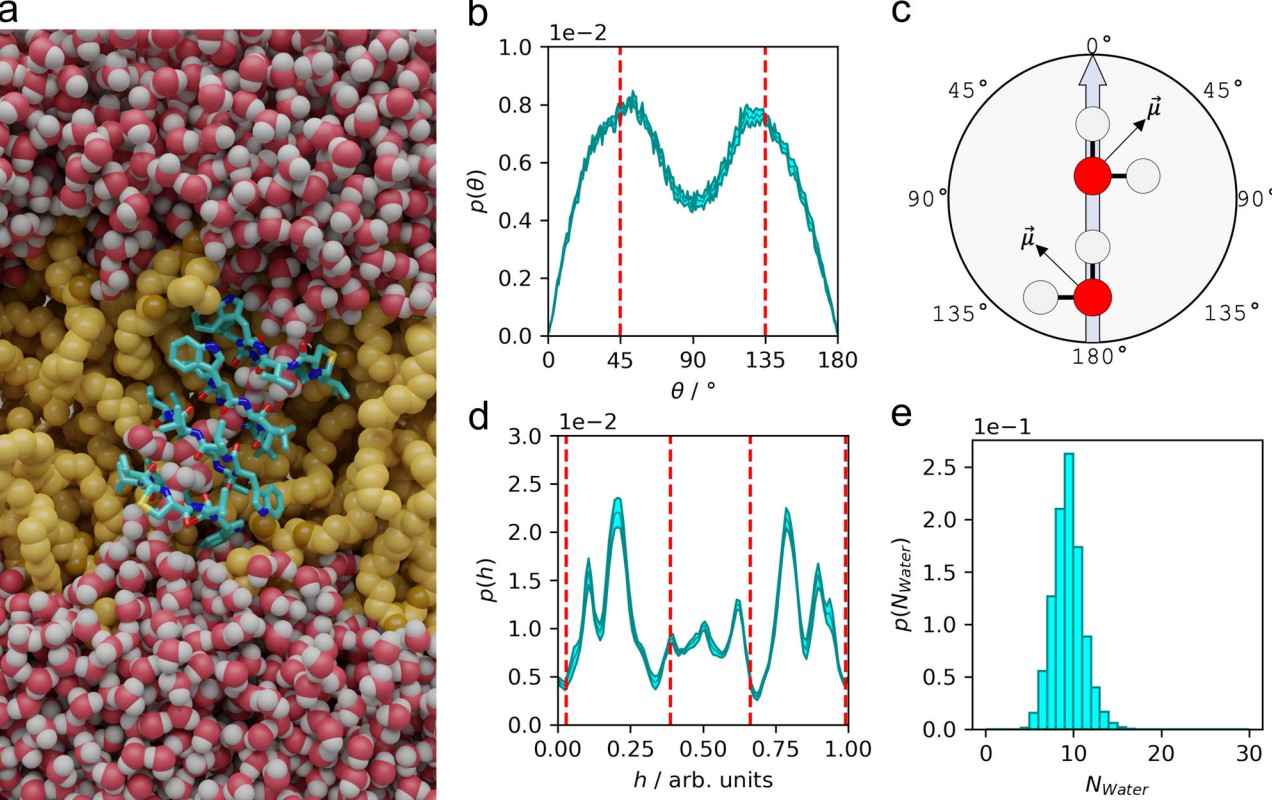

**Fig. 6 | Characteristics of water molecules within a lugdunin nanotube.** Data were taken from microsecond MD simulations of a nanotube consisting of four lugdunin molecules in *trans*-configuration embedded in a DOPC membrane. **a** Channel composed of four lugdunin peptides in tryptophan-*trans* configuration embedded in a DOPC membrane. **b**, **c** Dipoles of water molecules within the channel preferentially adopt an angle of 45° and 135° with the nanotube axis. **d** Density of water along the nanotube axis. **e** Number of water molecules within the nanotube.

closed water channel that connects both sides of the membrane (Fig. 6a, d) with, on average, ten water molecules within the channel (Fig. 6e). We observed a preferential orientation of the water dipoles within the channel at angles of 45° and 135° with the nanotube axis (Fig. 6b, c).

Our results provide a molecular picture of the active structure of the antimicrobial fibupeptide lugdunin in lipid membranes that explains the observation that lugdunin attacks membranes of Gram-positive bacteria without the need for protein components. The hydrogen-bonded water wire within the formed peptidic nanotubes allows the passage of protons thus dissipating the membrane potential of bacterial cells without forming large pores as observed for other AMPs.

**Single-channel properties of lugdunin nanotubes**
Our results derived from IR experiments and MD simulations strongly suggest that peptide nanotubes, formed by a lugdunin self-assembly in lipid membranes, comprise a water-filled membrane pore capable of transporting protons. For further support, single-channel proton conductance measurements would be desirable but cannot be performed at physiological conditions owing to orders of magnitudes too small proton concentrations. Therefore, further evidence for lugdunin's channel activity was envisioned by single-channel recordings in the presence of monovalent cations that are required to balance charge differences during proton transport. Channel recordings were pursued on black lipid membranes (BLMs) in a symmetric bathing solution composed of 500 mM MCl with M = $Na^+$, $K^+$, or $Cs^+$ and 10 mM HEPES, pH 7.4. Individual and stochastic channel events were recorded for each metal cation revealing several different conductance states (Fig. 7a, b, Supplementary Fig. 27). We

attribute these events to an underlying nanotube assembly/disassembly process leading to different channel architectures composed of a varying number of lugdunin subunits which affect the channel length and, thus, its conductive properties[40] (Fig. 7b, Supplementary Fig. 27). To compare the translocation of the employed metal cations, we determined the highest conductance state from Gaussian peak fitting to $G_{Na+} = 15.6$ pS $< G_{K+} = 29.3$ pS $< G_{Cs+} = 79.9$ pS. The conductance values follow the lyotropic series which is characteristic of membrane channels comprising a diffusive pore, like gramicidin A[57,58], further strengthening our hypothesis of a membrane-spanning lugdunin nanotube. Noteworthy, the obtained conductance value for potassium is close to that found for gramicidin A in glycerylmonooleate BLMs ($G_{K+} \approx 29$ pS)[57], suggesting that the effective pore size of the lugdunin nanotube is close to the 4 Å pore diameter of gramicidin A[59]. This assumption is in line with the diameter for lugdunin nanotubes determined to be 3.66 Å obtained from in silico studies using the HOLE program[60,61]. The determined single-channel dwell time constants were (479 ± 15) ms for $K^+$, (493 ± 34) ms for $Na^+$, and (705 ± 20) ms for $Cs^+$ (Fig. 7c, Supplementary Fig. 27). Generally, lugdunin proved to be an efficient transporter for all tested metal cations demonstrating that it does not only affect proton gradients in bacterial cells but also targets vital transmembrane gradients generated by a $Na^+$ or $K^+$ concentration imbalance[62,63].

In conclusion, our experimental results and molecular dynamics simulations draw a comprehensive picture of how lugdunin can dissipate the membrane potential of Gram-positive bacteria. Based on our results, we propose that lugdunin exerts its mode of action by a two-step process: First, lugdunin needs to efficiently partition into the hydrophobic core of the membrane. Second, transmembrane

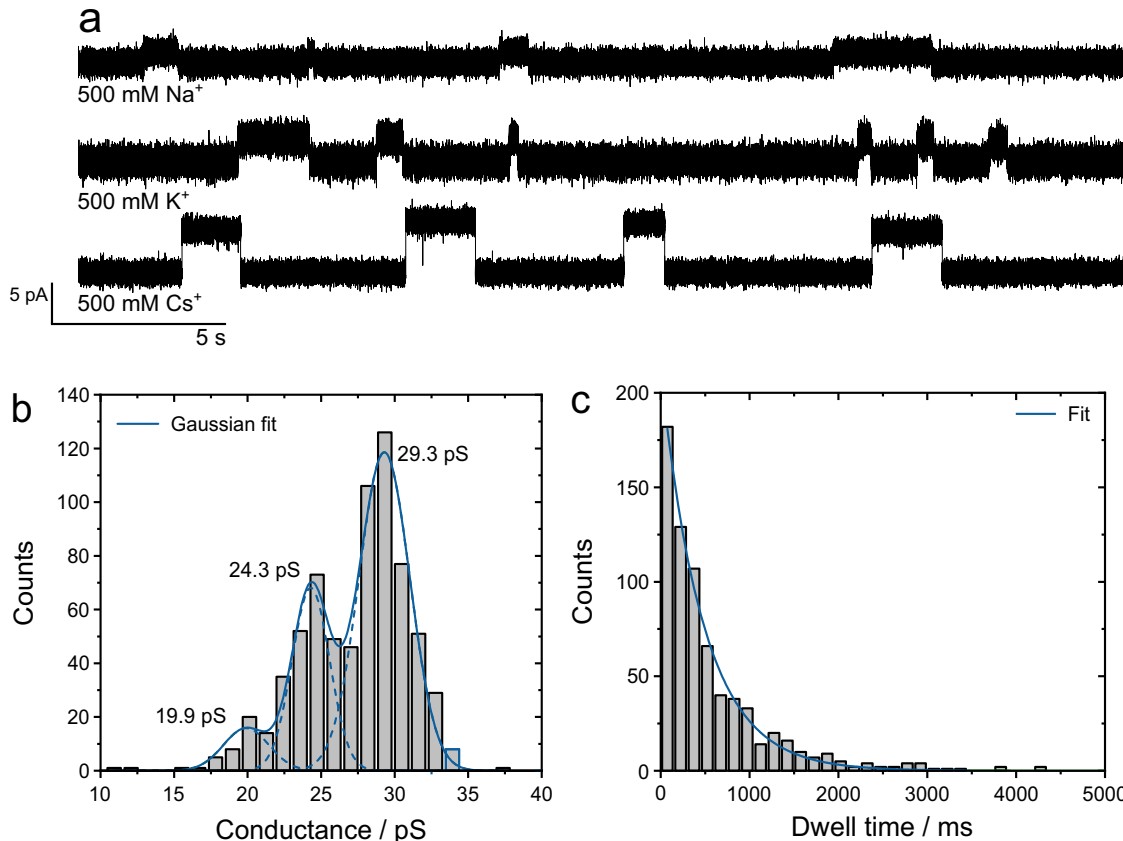

**Fig. 7 | Single-channel properties of membrane channels formed by lugdunin nanotubes.** **a** Recorded transmembrane currents across BLMs composed of POPC reveal single-channel conductance events in the presence of 500 mM Na$^+$, K$^+$, and Cs$^+$ at a voltage of +100 mV. **b** Event histogram ($n = 704$ events, bin width = 1.15 pS) of the K$^+$-conductance. Three Gaussian functions were fitted to the histogram revealing maximum conductance values of 19.9 pS, 24.3 pS and 29.3 pS. **c** Dwell time histogram of single-channel events in the presence of K$^+$ ($n = 701$ events, bin width = 150 mS, dwell times larger than 5000 ms were excluded). A mono-exponential fit yields a dwell time constant of $(479 \pm 15)$ ms.

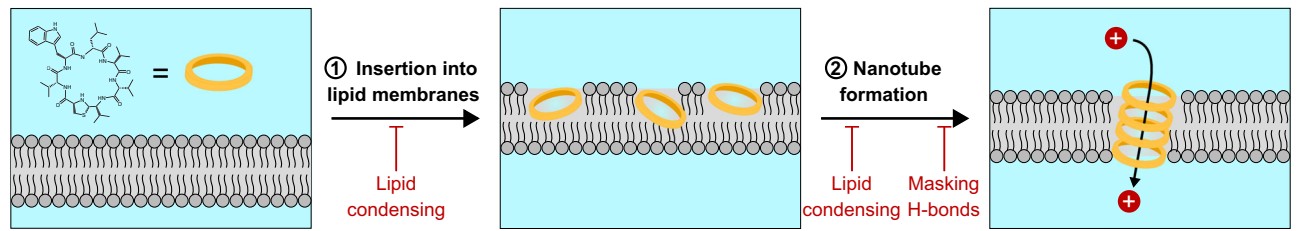

**Fig. 8 | Proposed two-step mechanism of lugdunin activity.** To exert its mechanism of action, lugdunin first inserts into lipid membranes and then self-assembles into peptide nanotubes via intermolecular hydrogen bonds. Both processes are vital for lugdunin's antibacterial activity and are affected by different membrane and peptide properties.

nanopores are formed by the spontaneous self-assembly of individual lugdunin monomers acting as membrane channels (Fig. 8).

This model is supported by multi-microsecond simulations of the interaction of the fibupeptide at high concentrations (peptide-to-lipid ratios between 1:40 and 1:10) with a Gram-positive membrane model: Following peptide insertion into the membrane, we observed the spontaneous formation of lugdunin nanotubes that traverse the membrane hydrophobic core (Supplementary Movie 1; trimers, tetramers, and a pentamer were observed). Consequently, these nanotubes allow for water passage across the membrane (Supplementary Movie 2).

The physical properties of the membrane greatly influence these two processes, the partitioning of lugdunin and the assembly process to form nanotubes, leading to the observed activity of the fibupeptide. I.e., cholesterol lipid-condensing and membrane thickening impair an efficient partitioning of the peptide but also the formation of hydrogen-bonded oligomers, while a *N*-methylation does not alter the partitioning of the peptide into lipid membranes but does interfere with its ability to participate in intermolecular hydrogen bonds preventing the formation of membrane-spanning nanotubes. The resulting conductive properties are governed by the size of the transporting ion but also by the self-assembly process whose dynamic nature enables an adaptation of the channel structure to the membrane properties. The proposed mode of action of the naturally occurring fibupeptide lugdunin carries implications for advancing the tailoring of antibiotics within this peptide class.

## Methods

### Solid-phase peptide synthesis of lugdunin 1 and 3,6-ditryptophan-lugdunin

The peptides were prepared by manual SPPS as described previously[48].

### General procedure for synthesis of N-methylated lugdunin analogs 3–5

**Solid-phase peptide synthesis.** For the synthesis of linear peptides a preloaded Fmoc-D-Trp(Boc) TentaGel S AC resin (loading: 0.23 mmol g$^{-1}$, 150 mg, 34.5 µmol, Rapp Polymere Tübingen) was used and swollen for 20 min in DMF (2 mL). Stepwise SPPS was performed in a 5 mL polypropylene syringe equipped with a 25 µm polyethylene frit purchased from Multisyntech GmbH (Witten, Germany) applying Fmoc strategy.

### Fmoc removal

The Fmoc protecting group was removed by treating the resin with the following washing solutions: 2,3,4,6,7,8,9,10-octahydropyrimido[1,2-a]azepine (DBU, 2%) and morpholine (10%) in DMF (2 mL) (1 × 12 min), DBU (2%) and morpholine (10%) in DMF (2 mL) (1 × 3 min), DMF (2 mL, 1 × 5 min), DCM (2 mL, 1 × 5 min) and DMF (2 mL, 1 × 5 min). The resin was then ready for the next coupling reaction.

### Coupling reaction

Coupling reactions were performed in DMF (2 mL) using 6 equiv. of the corresponding Fmoc-protected amino acids (Fmoc-L-Val-OH, Fmoc-D-Leu-OH, Fmoc-L-Trp(Boc)-OH, Fmoc-D-Val-OH, Fmoc-L-Thz(L-Val)-OH), 6 equiv. of HATU, 6 equiv. of HOBt and 8 equiv. of NMM. Coupling reactions were run for 45 min each while shaking at room temperature (Thermomixer 5437, Eppendorf). The reaction mixture was drained and the resin was washed with the following washing solutions: DMF (2 mL, 1 × 5 min), DCM (2 mL, 1 × 5 min), and DMF (2 mL, 1 × 5 min). The resin was subjected to Fmoc removal.

### On-resin N-methylation[54]

a) Introduction of 2-nitrobenzenesulfonyl activating group (oNBS): The resin-bound peptide with a free amine was washed with DCM (2 mL, 3 × 5 min) and treated with a mixture of oNBSCl (4 equiv.) and 2,4,6-trimethylpyridine (sym-Collidine, 5 equiv.) in DCM (2 mL) for 1.5 h while shaking at room temperature. The reaction mixture was drained and the resin was washed with the following washing solutions: DCM (2 mL, 3 × 5 min) and DMF (2 mL, 3 × 5 min). b) N-methylation: Selective N-methylation was achieved by treating the N-oNBS protected on resin-bound peptide with a mixture of dimethyl sulfate (DMS, 3 equiv.) and 7-methyl-1,5,7-triazabicyclo(4.4.0)dec-5-ene (mTBD, 3 equiv.) in DMF (2 mL) for 30 min while shaking at room temperature. The reaction mixture was drained and the resin was washed with DMF (2 mL, 3 × 5 min). c) oNBS removal: Removal of oNBS moiety was achieved by treating the resin with a mixture of DBU (5 equiv.) and 2-mercaptoethanol (10 equiv.) in DMF (2 mL) for 30 min while shaking at room temperature. The reaction mixture was drained and the resin was washed with DMF (2 mL, 3 × 5 min). The resin was then ready for the next coupling reaction, which was performed two times (1 × 3 h, 1 × overnight) for more efficient coupling.

### Resin cleavage of linear peptide

Once the desired peptide was generated, the final Fmoc protecting group was removed following Fmoc removal procedure with the following additional washings: DMF (2 mL, 3 × 5 min), DCM (2 mL, 3 × 5 min), toluene (2 mL, 3 × 5 min), isopropyl alcohol (2 mL, 3 × 5 min), and diethyl ether (2 mL, 3 × 5 min). The resin was dried in vacuo for 2 h and cleaved from the linear peptide using a mixture of trifluoroacetic acid (TFA), triisopropylsilane (TIPS), and $H_2O$ (95/5/5, v/v/v, 2 mL, 3 × 1 h). The filtrate was concentrated in vacuo and the resulting residue was dissolved in a mixture of tert-Butyl alcohol and water (1/1, v/v), frozen and lyophilized.

### Macrolactamization

The linear N-methylated heptapeptide was applied to macrolactamization under high dilution conditions (2 mM) in DMF using HATU (4 equiv.), 3-hydroxytriazolo[4,5-b]pyridine (HOAt, 6 equiv.), N-ethyl-N-(propane-2-yl)propane-2-amine (DIPEA, 8 equiv.) as coupling reagents for 24 h. The yellow solution was then diluted with 20 mL $H_2O$ and extracted one time with an n-BuOH/CHCl$_3$ (1/5, v/v) solution and two times with CHCl$_3$. The joint organic phases were washed three times with an aqueous 10% KHSO$_4$ solution, three times with a saturated aqueous NaHCO$_3$ solution, a saturated aqueous NaCl solution, and an aqueous 15% acetonitrile solution. The solvent was removed under vacuum and the residue was taken up in a mixture of tert-butyl alcohol and water (1/1, v/v), frozen, and lyophilized to obtain the product as a white voluminous powder, which was then applied to HPLC to get pure selectively N-methylated peptides (yield: 28% to 53%).

### Purification and HPLC-analysis of N-methylated lugdunin analogs (3–5)

Purification was performed with 77% MeOH with 0.1% formic acid (FA), and 23% water with 0.1% formic acid for 30 min on an RP (reversed phase)-HPLC column (Kromasil 100 C18, 7 µm, 250 × 4 mm, Dr. Maisch GmbH) with a flow rate of 13.5 mL min$^{-1}$. Eluents were monitored at 220 nm, collected, and freeze-dried. Purity was determined by LC-UV-HRMS (high-resolution mass spectrometry, ESI(+)-TOF-instrument, MaXis-4G HD, Bruker GmbH; Bruker Compass DataAnalysis V. 4.2 SR2) and proton NMR analysis (MestReNova V. 8.1.-12489, HPLC-UV-MS chromatograms are displayed in the supporting information). All experimental results are given in Supplementary Figs. 1–9.

### Vesicle preparation

A lipid film was obtained by drying the desired amount of a lipid solution in CHCl$_3$ under a gentle stream of $N_2$ at 30 °C. For tryptophan fluorescence experiments, lipid films composed of varying amounts of POPC, POPG, and cholesterol (Avanti Polar Lipids, Alabaster, AL, US) with a mass of 8 mg were prepared, while for ATR-FTIR experiments lipid films composed of DMPC and a mass of 2 mg were used. Residual solvent was removed in vacuo for at least 3 h. For fluorescence experiments, a lipid film was rehydrated with 1 mL HEPES buffer (100 mM KCl, 10 mM HEPES, pH 7.4) for 30 min and then vortexed 3× for 30 s in 5 min intervals. For ATR-FTIR measurements, 500 µL ultrapure water was used. Large unilamellar vesicles (LUVs) were produced by extruding the lipid suspension 31× through a polycarbonate membrane with a nominal pore diameter of 200 nm (LiposoFast-Basic, Avestin, Ottawa, Canada). In the case of DMPC, the preparation was conducted at 30 °C accounting for its main phase transition temperature of 24 °C. The final lipid concentration was determined by quantifying the inorganic phosphate content[12].

### Tryptophan fluorescence

All tryptophan fluorescence spectroscopic measurements were conducted with a FluoroMax-4 (Horiba, Kyoto, Japan; FluorEssence V. 3.1.5.11) at 22 °C and a 10 mm × 4 mm quartz cuvette (Hellma, Müllheim, Germany) under constant stirring. Tryptophan fluorescence was excited at $\lambda_{max}$ = 280 nm and recorded between 300-400 nm by averaging 25–40 spectra. Cross-orientated polarizers were used to reduce the influence of scattered light (Pol$_{Ex}$ = 0°, Pol$_{Em}$ = 90°).

## Partition coefficient

The partition coefficient $K_\chi$ of a molecule M between an aqueous buffer and a lipid membrane is defined as:

$$K_\chi = \frac{\chi_{M,membrane}}{\chi_{M,aqueous\ solution}} \tag{1}$$

where $\chi_M$ refers to the mole fraction of the substance in the aqueous or the lipid phase, respectively. To determine a partition coefficient, a 10 μM lugdunin solution in HEPES buffer was titrated with a vesicle suspension of the desired lipid composition. Fluorescence spectra were recorded in the absence and in the presence of 10 μM, 25 μM, 50 μM, 100 μM, 250 μM, 500 μM, 750 μM, 1000 μM, 1500 μM, and 2000 μM lipid. After each addition of vesicles, the suspension was incubated for 10 min to achieve equilibrium. The spectra were background corrected and further scattering effects were removed with a reference titration using 10 μM tryptophan instead of lugdunin as described by Ladokhin et al.[22]:

$$I_{corrected} = I \frac{I_{0,Trp}}{I_{Trp}} \tag{2}$$

Here, $I$ and $I_{Trp}$ are the fluorescence intensities of lugdunin and tryptophan, respectively. $I_0$ always denotes the measured intensity in pure buffer. The partition coefficient $K_\chi$ was obtained from fitting Eq. (3) to the normalized corrected fluorescence intensities dependent on the lipid concentration:

$$\frac{I_{corrected}}{I_{0,corrected}} = 1 + (I_\infty - 1) \cdot \frac{K_\chi \cdot [L]}{[W] + K_\chi \cdot [L]} \tag{3}$$

$I_\infty$ is the fluorescence intensity upon full binding and [L] is the lipid concentration. [W] denotes the molar concentration of water with a value of 55.3 M.

## Quenching experiments

Collisional quenching of lugdunin fluorescence was measured in the presence of different concentrations of iodide ions by using a 4 M KI stock solution to which 1 mM of $Na_2S_2O_3$ was added to prevent the formation of polyiodide ions. With a 4 M KCl solution, the additional ionic strength of the solution was adjusted to 150 mM in every experiment. A 5 μM lugdunin solution in HEPES buffer was used. The measurements were either conducted in the absence of vesicles or with a peptide-to-lipid ratio of 1:100 ($n/n$) corresponding to a lipid concentration of 500 μM. All spectra were background corrected and the data was analyzed using the Stern–Volmer equation (Eq. (4)):

$$\frac{I_0}{I} = K_{SV} \cdot [I^-] + 1 \tag{4}$$

$I_0$ and $I$ are the measured intensities at $\lambda = 350$ nm in the absence or presence of quencher molecules, respectively. $[I^-]$ denotes the iodide concentration and $K_{SV}$ the Stern–Volmer constant which is a measure of the accessibility of the tryptophan residue for quencher molecules.

## Proton transport assays

Proton transport induced by lugdunin and its analogs was analyzed using the pH-sensitive dye pyranine (8-hydroxypyrene-1,3,6-trisulfonic acid). LUVs were filled with 100 mM KCl, 10 mM HEPES, and 0.5 mM pyranine (pH = 7.4), and extravesicular dye was removed after extrusion via size exclusion chromatography (Illustra NAP-25 G25, GE Healthcare, Chalfont St Giles, UK). The vesicles were diluted in the same buffer without pyranine and pH = 6.4 to a final lipid concentration of 50 μM. Pyranine fluorescence was monitored in a time-

dependent manner with $\lambda_{ex} = 458$ nm, $\lambda_{em} = 512$ nm, and band widths of 3 nm using an FP 6500 spectrofluorometer (Jasco Germany, Groß-Umstadt, Germany; Spectra Manager V. 1.54.03) under constant stirring. After acquiring a baseline for 100 s, peptide stock solution in isopropanol was added to a nominal peptide-to-lipid ratio ($n/n$) of 1:250. Acidification of the lumen resulted in fluorescent quenching and was monitored over the course of 500 s. Afterward, the vesicles were lysed by the addition of N,N-dimethyl-n-dodecylamine N-oxide (LDAO) leading to a disruption of the pH gradient. All data points were normalized to the fluorescence intensity directly before the addition of the peptide and after vesicle lysis.

## Chloride ion transport assay

The potential transport of chloride ions by lugdunin was assessed using a lucigenin-based vesicle assay. LUVs were prepared as described above and filled with 225 mM $KNO_3$, 10 mM HEPES, and 0.8 mM lucigenin (pH 7.4). Free dye molecules were removed via size exclusion chromatography (Illustra NAP-25 G25, GE Healthcare, Chalfont St Giles, UK). Before the experiment, vesicles were diluted to a volume of 800 μL and a lipid concentration of 50 μM and incubated for 3 min in the presence of 0.2 μM lugdunin under constant stirring (peptide-to-lipid ratio of 1:250). Lucigenin fluorescence was excited at $\lambda_{ex} = 430$ nm and monitored at $\lambda_{em} = 505$ nm with 3 nm slit widths. A baseline was recorded for 100 s, before the chloride gradient was established by the addition of 10 mM KCl to the cuvette from a 1 M stock solution. Changes in fluorescence intensity were monitored over the course of 10 min before the addition of LDAO resulted in vesicle lysis and the disruption of the chloride gradient. All data were normalized to the fluorescence intensity directly before the establishment of the chloride gradient and after vesicle lysis. For control experiments without lugdunin, pure isopropanol was used instead.

## ATR-FTIR experiments

24 h before and during each measurement, the IR spectrometer (Vertex70, Bruker, Ettlingen, Germany; OPUS V. 6.5) was extensively purged with dry air to remove any atmospheric water and $CO_2$ from the system. A 45° Ge-crystal with six internal total reflection points was used as an internal reflection element (IRE). Before mounting it on the horizontal ATR-unit (A537-A/Q, Bruker, Ettlingen, Germany), it was cleaned with Hellmanex (Hellma, Müllheim, Germany) and methanol. After recording the respective background spectra of the clean crystal for each polarization (unpolarized, perpendicular, or parallel polarized incident light was adjusted with wire grid polarizers (Optometrics Corporation, Ayer, US)), at least 500 μg of the sample were deposited onto the crystal and dried for about 3 h until no change in the IR spectrum was observed. Pure DMPC vesicles or lugdunin-containing DMPC vesicles were used. The latter were obtained by incubating lugdunin and DMPC vesicles in the desired nominal peptide-to-lipid ratio ($n/n$) for 30 min at 30 °C, followed by a size-exclusion chromatography (Illustra NAP-10 G25, GE Healthcare, Chalfont St Giles, UK) to remove unbound peptide molecules. After the sample was dried, final spectra were recorded with a room temperature deuterated lanthanum α-alanine-doped triglycine sulfate (RT-DLaTGS) detector, a resolution of 4 cm$^{-1}$, and an average of 120 scans. After each measurement, the IRE was cleaned with ultrapure water and isopropanol. Each spectrum was ATR-corrected, smoothed with nine smoothing points, and baseline corrected in the range of the amide I band (1700 cm$^{-1}$ < ṽ < 1600 cm$^{-1}$) using the OPUS software (Bruker, Ettlingen, Germany). All data points were normalized to the highest absorbance peak, i.e., the asymmetric stretching vibration of the methylene groups. Exact peak positions were determined from the second derivative spectrum and peak deconvolution was performed via the peakfit function in MATLAB. For this, a Lorentzian peak shape was applied.

## Calculation of orientational properties

The spectra obtained with perpendicular or parallel polarized light, respectively, were used to determine the orientation of the sample on the IRE. The detailed experimental setup is depicted in Supplementary Fig. 10. The values of the electric field of the polarized incident light ($E_{\parallel}$ or $E_{\perp}$) have certain contributions to the spatial electric field amplitudes $E_x$, $E_y$, and $E_z$ of the resulting evanescent waves at the internal total reflection points. $E_{\parallel}$ contributes to $E_x$ and $E_z$, whereas $E_{\perp}$ only influences $E_y$. The electric field amplitudes $E_x$, $E_y$, and $E_z$ were calculated as follows[64]:

$$E_x = \frac{2\cos(\gamma)\sqrt{\sin^2(\gamma) - n_{31}^2}}{\sqrt{1 - n_{31}^2}\sqrt{(1 + n_{32}^2)\sin^2(\gamma) - n_{31}^2}}, \tag{5}$$

$$E_y = \frac{2\cos(\gamma)}{\sqrt{1 - n_{31}^2}}, \tag{6}$$

$$E_z = \frac{2\cos(\gamma)n_{32}^2\sin(\gamma)}{\sqrt{1 - n_{31}^2}\sqrt{(1 + n_{32}^2)\sin^2(\gamma)n_{31}^2}} \tag{7}$$

Here, $\gamma$ refers to the incident angle of the IR beam with a value of 45° and $n$ denotes the refractive indices of the Ge-crystal ($n_1$), the sample material ($n_2$) and the bulk medium above the sample ($n_3$), respectively. The notations $n_{31}$ and $n_{32}$ denote the quotients of the individual refractive indices. As the penetration depth of the evanescent waves in our setup does not pass the sample completely, the thick film approximation can be applied[64]. Thus, in Eqs. (5), (6) and (7) the bulk medium does not have to be considered and $n_3$ equals $n_2$. Knowing the values of the electric field components allows the determination of the order parameter $S(\Theta)$ defined as:

$$S(\Theta) = \frac{2\left(E_x^2 - R^{ATR}E_y^2 + E_z^2\right)}{(3\cos^2(\alpha) - 1)\left(E_x^2 - R^{ATR}E_y^2 - 2E_z^2\right)} = \frac{1}{2}(3\cos^2(\Theta) - 1) \tag{8}$$

$\alpha$ describes the angle of the transition dipole moment of the vibrational mode that is investigated. In the case of lipid acyl chains, this angle is 90°. For lugdunin we assume the formation of antiparallel β-sheet-like peptide nanotubes. In these nanotubes, the backbone carbonyl and N-H-bonds are oriented parallel to the central tube axis. Thus, the transition dipole moment of the perpendicular component of the amide I band and the tube axis coincide. With these assumptions, $\alpha$ becomes 0°. $R^{ATR}$ depicts the dichroic ratio of the same vibrational mode and can be calculated from the peak integrals obtained from parallel ($A_{\parallel}$) and perpendicular ($A_{\perp}$) polarized spectra:

$$R^{ATR} = \frac{A_{\parallel}}{A_{\perp}} \tag{9}$$

For lipid acyl chains the symmetric and antisymmetric stretching modes of the $CH_2$-groups were used, while for the amide I mode the peak at around 1640 cm$^{-1}$ was taken. By calculating the dichroic ratio and the values of the electric field components, one can determine the order parameter and with this, the effective angle $\Theta$ describing the angle between the molecular axis and the surface normal of the IRE. All values used in our analysis are summarized in Supplementary Table 1.

## Voltage-clamp experiments

Electrical measurements were performed using a 16-channel microelectrode cavity array (MECA, Ionera Technologies GmbH, Freiburg, Germany) interfaced to the Orbit16 (Nanion Technologies, Munich, Germany). Lipid solutions used for painting of black lipid membranes

(BLMs) consisted of POPC dissolved in $n$-octane to a concentration of 5 mg mL$^{-1}$. For the preparation of BLMs, 200 µL of the bathing solution (500 mM MCl (with M = K, Na, Cs), 10 mM HEPES, pH 7.4) was added to the measurement chamber and the current was measured using Ag/AgCl electrodes and the software Elements Data Reader at a sampling rate of 20 kHz. 0.1–0.3 µL of lipid solution were added in close proximity to the microcavities, followed by painting of lipid bilayers by gentle rotation of a magnetic stirring bar. A transmembrane voltage of +100 mV was applied. Lugdunin was added in steps of 2.5 nM to a maximum concentration of 10 nM until single-channel conductance events were observable. Recorded current traces were filtered with a Bessel 8 pole lowpass 1000 Hz filter and evaluated with the single-channel search implemented in Clampfit 11.2.2.17 extracting the event amplitude and dwell time. All extracted events were manually accepted and the conductances were obtained from the current amplitude divided by the transmembrane voltage of +100 mV. Event conductances and dwell times were plotted as histograms with bin widths adjusted according to the Freedman-Diaconis rule. Gaussian functions were fitted to the conductance histograms using the peak fit routine in MATLAB, whereas a monoexponential function was fitted to the dwell time histograms yielding the dwell time constant.

## Parameterization of thiazolidine moiety

The thiazolidine moiety was parameterized in this study using the CHARMM36m protein force field[65,66] and the CHARMM General Force Field version 4.6 (CGenFF)[67]. The potential energy function of the nonpolarizable all-atom CHARMM force field was already thoroughly described elsewhere in the literature[68,69]. Atom types, partial atomic charges, and missing bonded parameters were assigned using the ParamChem web server (program version 2.5)[70–72]. Partial atomic charges or bonded parameters not already optimized in the CHARMM36m protein force field or CGenFF were refined. Lennard-Jones parameters of the assigned atom types were not optimized. Molecular mechanical (MM) calculations were carried out using NAMD2[73] and GROMACS 2021.7[74–76], while quantum mechanical (QM) calculations were performed using GAUSSIAN09[77].

The parameterization of the thiazolidine moiety was split in two stages. Partial atomic charges and the main part of the missing bonded parameters for the CGenFF force field were parameterized using molecule A (Supplementary Fig. 11a). Bonded terms including both CHARMM36m protein and CGenFF atom types were parameterized based on molecule B (Supplementary Fig. 11b). The general workflow used for parameterization of both molecules is illustrated in Supplementary Fig. 12. The optimization process was iterated twice to ensure self-consistency of the force field.

The initial coordinates for the thiazolidine moiety were obtained from the NP-MRD database[78] (ID: NP0015612, last accessed: 14.04.2023). For the structure optimization, the MP2/6-31G(d) model chemistry was employed. The optimized structure was subsequently used to refine the partial atomic charges in the MM model using the approach outlined by Croituro et al.[69] Ab initio minimum interaction energies were calculated for water-model complexes at the HF/6-31G(d) level for non-sulfur atoms. Water molecules were in TIP3P model configuration and were set in linear orientation to the hydrogen bond acceptor- or donor atom. The interaction axis was fixed during the QM calculation and multiple rotations of the water molecule around the axis were considered in the case of polar atoms. For all polar interactions, the ab initio minimum interaction energies were scaled by an empirical factor of 1.16, and the minimum interaction distance was corrected by subtracting 0.2 Å[68]. The MP2/6-31G(d) model chemistry was used for the sulfur atom considering the basis set superposition error (BSSE) correction of Boys and Bernardi without applying any scaling or offset rules[79]. Additionally, the molecular dipole moment was calculated at the MP2/6-31G(d) level in vacuum and was used as target data in the charge fitting process. The ab initio

dipole moment target was increased by 30% to elevate the molecular polarizability of the MM model according to the standard CHARMM protocol[67,69]. To improve the charge fitting for atoms that do not directly participate in hydrogen bonds, the electrostatic potential (ESP) of the molecule was calculated at the MP2/6-31G(d) level and was also added to the objective function of the optimization process.

The final objective function reads:

$$f_{\text{Obj}} = w_{\text{E}} \cdot \sqrt{\frac{1}{N_{\text{A}}} \cdot \sum_{i=0}^{N_{\text{A}}} (E_i^{\text{QM}} - E_i^{\text{MM}})^2} + w_{\text{Dipole}} \cdot \|1.3 \cdot \vec{d}_{\text{QM}} - \vec{d}_{\text{MM}}\|_2$$
$$+ w_{\text{ESP}} \cdot \sqrt{\frac{1}{N_{\text{P}}} \cdot \sum_{j=0}^{N_{\text{P}}} (ESP_j^{\text{QM}} - ESP_j^{\text{QM}})^2}$$

(10)

The first term is the root-mean-square deviation (RMSD) between the ab initio minimum energies and the MM force field, the second term is the 2-norm between the scaled QM dipole moment and the dipole moment of the MM model, and the last term is the RMSD between the QM and MM electrostatic potential (ESP) at each evaluated point in space. The weight for the energy term was set to 10 kcal$^{-1}$mol, the weight for the dipole contribution to 3.0 D$^{-1}$ (D = Debye), and the weight for the ESP to 1 kcal$^{-1}$mol. The angle between the QM and MM dipole moment, as well as the minimum interaction distance, were not explicitly included in the charge fitting process but were utilized as external validation metrics for the fit result.

The optimization of bonded parameters, including bonds and angles, was carried out following the workflow outlined by Mayne et al.[80], which involved the use of 3-point potential energy surface (PES) scans. In order to introduce small distortions to the molecular structure, an internal coordinate (IC) representation was employed. The resulting energy differences between the distorted and undistorted structures in the MM model were calculated using the NAMD2 program[73]. QM energy differences were obtained based on a Hessian matrix calculated at the MP2/6-31G(d) level of theory and scaled by a factor of 0.89[80]. During the optimization process, the squared difference between the ICs of the bond and angles, as well as the squared differences of the energies were minimized. The structural difference term was given twice the weight of the energy difference term in this study. All steps were performed using the ffTK software package[80].

Bonded interactions involving four atoms are described by proper dihedrals in the CHARMM force field[68]. ffTK provides a suitable suite to optimize parameters for missing dihedral terms based on the initial guesses of the ParamChem webserver. It uses thereby QM derived torsion profiles calculated at the MP2/6-31G(d) level of theory[80]. The phase values for proper dihedral terms were assigned by the ParamChem webserver[70-72]. The dihedral multiplicities were adjusted to fit the QM and MM torsion profiles without excessive overfitting of the parameters. Additional information about the parameterization process for bonded parameters is available in the corresponding publication of ffTK[80].

Supplementary Table 2 shows probe sites used for charge fitting, the angle of the water molecule relative to the interaction axis, scaled QM energies and shifted QM distances with corresponding initial and optimized MM values for both optimization iterations. For the second optimization round, the energy-minimized structure of the MM model from the first optimization round was used to calculate a new set of water-model complexes. Supplementary Table 3 contains the optimal and initial values in each optimization round for the objective function, the RMSD of the energies, the RMSD of the distance, the angle between the QM and MM dipoles, and the RMSD of the ESP. Optimal and initial partial atomic charges of the atoms in the thiazolidine moiety are provided in Supplementary Table 4. Supplementary Table 5 shows the results of the bonded parameter optimization for both the first and

second iteration rounds, while Supplementary Fig. 13 displays the QM and MM torsional profiles of the optimized dihedrals. The QM Hessian matrix and torsional profiles, calculated for the QM-optimized structure in the first iteration, were re-used in the second optimization iteration.

The residue was extended in the second stage in N-terminal direction to account for all necessary combinations of CGenFF and CHARMM36 atom types (Supplementary Fig. 11b). For the L-valine residue adjacent to the thiazolidine moiety, atom types and partial charges were obtained from the CHARMM36 protein force field. Partial atomic charges and optimized CGenFF parameters for the thiazolidine ring were transferred from molecule A without further refinement. Parameters, including CHARMM36 and CGenFF atom types, were either adopted from Croitoru et al.[69] or derived by analogy from CGenFF, and were then refined accordingly. Angle potentials including an additional Urey-Bradley term (CT1-CT1-CG3C51) were not considered for optimization. Supplementary Table 6 and Supplementary Fig. 14 summarize the results of the bonded parameter optimization for molecule B. After the second iteration of the parameterization workflow, the structure was optimized using the improved MM model. Supplementary Fig. 15 illustrates the alignment between the QM and MM models, together with the corresponding RMSD values.

The topology of the peptide was built using the pdb2gmx utility of GROMACS. The CHARMM36m force field was used to represent standard L- and D-amino acids. However, since the backbone of the thiazolidine moiety is not described in the CHARMM36m force field, only the CMAP term from the D-valine to the L-valine residue of the thiazolidine moiety was directly included. More information regarding the parameters can be found in the associated Zenodo repository[81]. Lipid parameters were taken from the CHARMM36 force field. The recommended TIP3P water model was used for all simulations[68]. Upon acceptance of the paper, the parameters will be made available via a Zenodo repository.

## General parameters for MD simulations

All simulations (Supplementary Table 7) were performed using either the 2021.6 or 2021.7 version of GROMACS with a time step of $\Delta t = 2$ fs[74,75,82]. Pair-lists were generated using the Verlet cutoff scheme with a Verlet-buffer-tolerance of 0.005 kJ mol$^{-1}$ ps$^{-1}$, and the Lennard-Jones potential was shifted to zero between 1.0 nm and 1.2 nm. Electrostatic interactions were treated with the Particle-Mesh-Ewald (PME) algorithm with a cutoff of 1.2 nm[83]. Temperature and pressure were maintained using the v-rescale thermostat[84] and the Parrinello-Rahman barostat[85] with coupling constants of $\tau_{\text{T}} = 0.1$ ps and $\tau_{\text{P}} = 5.0$ ps, respectively. The system compressibility was set at 4.5e−5 bar$^{-1}$, and the reference pressure was kept at 1 bar for all simulations. Temperature was kept at 298.15 K for all simulations, unless otherwise specified. Periodic boundary conditions were applied for all simulations. All bonds including hydrogen atoms were constrained and handled with the LINCS algorithm[86]. Structures were prepared by an energy minimization (maximal 10,000 steps) deploying the steepest descent algorithm with a maximum step size of 0.0001, followed by equilibration simulations using the Berendsen barostat[87] together with a time step of $\Delta t = 1$ fs.

## Potential of mean force (PMF)

Potential of mean force (PMF) profiles for the insertion of ludgunin (1) and the derivative (lugdunin$^{\text{Cys4Ala}}$) into a lipid bilayer were obtained using Umbrella Sampling (US) simulations[88,89] (PMF curves displayed in Fig. 1e, f). The initial structures for US simulations, both in the aqueous phase and at the membrane-solvent interface, were derived from unbiased MD simulations (Supplementary Table 7). For peptides inserted into the hydrophobic core of the membrane, the center of mass (COM) of the peptide backbone was pulled along the z-axis toward the membrane core, employing a pulling rate of 0.0001 nm/ps

and a force constant of 1000 kJ/mol/nm². The spatial distance between the initial US structures ranged between 0.05 nm and 0.1 nm, with a finer resolution near the membrane core (histograms of the umbrella simulations shown in Supplementary Fig. 19).

Each US window was shortly equilibrated for 100 ps and was then simulated for 1 μs. Results were converged for this timescale (Supplementary Fig. 20). The relative distance between the peptide backbone COM and the membrane COM was restrained along the z-axis with a force constant of 1000 kJ/mol/nm², and for a subset of windows, a higher force constant up to 4000 kJ/mol/nm² was applied. The reference temperature was set at 298.15 K with a reference pressure of 1 bar. Remaining MD parameters were selected as described above.

The PMF profiles were computed using the Weighted Histogram Analysis Method (WHAM) integrated into the GROMACS software package. For all simulations, data from the last 990 ns were taken into account. Supplementary Fig. 19 illustrates the histograms of the relative distances used for the PMF calculation.

### In silico partition coefficients

The in silico partition coefficient for lugdunin is defined by the ratio of its concentrations in the lipid phase (thickness $h$) and in the aqueous phase (Eq. (11)). It is determined from the PMF, i.e., the Gibbs free energy $G(z)$ along a reaction coordinate $z$ according to Ghysels et al.[90]

$$K_{\chi,\text{sim}} = \frac{1}{h} \int_h^0 e^{-\beta(G(z)-G_{\text{ref}})} dz \qquad (11)$$

Here, $G_{\text{ref}}$ is the value for the free energy of lugdunin in the water phase (constant), $\beta = (k_B T)^{-1}$, $T$ the reference temperature, and $h$ denotes the outer interaction radius of the membrane for contact to lugdunin. 95% confidence intervals were computed by bootstrapping complete histograms (utilizing the hist option of gmx wham) for each system. Expectation values and quantiles of the profiles and the partition coefficients were estimated based on 500 bootstrapped profiles.

### Reporting summary

Further information on research design is available in the Nature Portfolio Reporting Summary linked to this article.

## Data availability

The data that support this study are available from the corresponding authors upon request. Fluorescence data and voltage clamp current traces used in this study are deposited at gro.data [https://doi.org/10.25625/8P4KJC][91]. All simulation models, input files and structures are available for download from Zenodo [https://doi.org/10.5281/zenodo.10839126][81]. Source data are provided with this paper.

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

## Acknowledgements

The authors thank Jutta Gerber-Nolte and José-Manuel Betran-Beleña for technical support. C.S. thanks the Deutsche Forschungsgemeinschaft (CRC 1456, project B04) for financial support. Work of S.N.W., T.D. and S.G. is financed by grants from the Cluster of Excellence EXC2124 Controlling Microbes to Fight Infection (CMFI, project ID 390838134). M.F.W.T. and R.A.B. gratefully acknowledge the compute resources and support provided by the Erlangen Regional Computing Center (RRZE) and the Erlangen National High Performance Computing Center (NHR@FAU). Special thanks are owed to Alexey Aleksandrov and the colleagues at the Computer Chemistry Center (CCC) at FAU for their invaluable discussions and insights on the parametrization of the thiazolidine moiety. We acknowledge support by the Open Access Publication Funds/transformative agreements of the Göttingen University.

## Author contributions

D.R. designed and coordinated the study, optimized and performed in vitro assays and wrote the manuscript. H.F. conducted parts of the fluorescence and IR-spectroscopic measurements. S.M. conducted parts of the fluorescence spectroscopic measurements. M.F.W.T. parameterized thiazolidine for simulation, setup and performed all simulations and analysis. R.A.B. acquired funding, designed and coordinated the simulation study. S.N.W. synthesized lugdunin and T.D. *N*-methyl-lugdunin analogs. S.G. contributed to the design of the study, the writing of the manuscript and acquired funding. C.S. acquired funding, coordinated the study and wrote the manuscript. All authors contributed in analyzing and discussing data and commenting on the manuscript.

## Funding

## Competing interests

Eberhard Karls Universität Tübingen holds a patent for lugdunin (EP 3072899B1) as inventors. The authors declare no competing interests.
