## [Peer Review File · Nature Communications]

The antimicrobial fibupeptide lugdunin forms water-filled channel structures in lipid membranes

Editorial Note: Parts of this Peer Review File have been redacted as indicated to remove third party material where no permission to publish were obtainedReviewer #1 (Remarks to the Author):

The manuscript describe the work carried out to understand the mechanism of action of antimicrobial peptide lugdunum. This is a natural cyclic peptide with antimicrobial properties that was discovered few years ago (Nature 2016m 535, 511). The lack of new antimicrobial drugs is creating a main concern to health authorities. There are a number of antibiotic-resistant bacterial strains whose treatment is difficult. Therefore the development of new antimicrobial with different mechanism of action are required. At this respect this cyclic peptide represent a very promising candidate. Since the first publication a number of works have already study the mechanism of action of this AMP, finding the proton translocation as one of the main consequences of dissipation of the membrane potential. The authors try to evaluate this mechanism using artificial models. They propose a channel-type structure as responsible of the proton transfer based on the computational studies. Other studies, such as FTIR support the antiparallel-like structure of the cyclic peptide.

Regarding the computational studies, how many hydrogen ions support the proposed tubular structure? although FTIR suggest the antiparallel stacking, did the author check the stability of this beta-sheet. How many water molecules are filling the channel of the tubular structure? Why the channel is made only by four cyclic peptides? did the author evaluated five or six subunits? in fact, Ghadiri work with D,L-cyclic peptides already reported that the most commonly observed species are the five stacked subunits (Acc. Chem. Res 2013, 46, 2955).

To carry the studies in transport they have used N-methylated peptides following Ghadiri precedent, I would recommend to cite one of his work as pretenders (ref 42 in the manuscript that is only related with nanotube forming cyclic peptide (Nature 1994, 366, 324)). I would also recommend to mention with more detail the synthetic procedure used to prepare this peptide. Also add the reference about the N-methylation on the resin. This si not a so general method. In the supporting information experimental data for the preparation fo the peptides should be provided.

Although the lugdunin has a similar structure to Ghadiri D,L- cyclic peptides or also to Mannopectimycins (that is also a natural alternating D,L-cyclic peptide). the transport mechanism can be different. Did the author checked if they are able to transport chloride? I would recommend to carry lucigenin transport experiments (halide sensor dye) to discharge that the peptides are not transporting HCl through a different mechanism but with similar consequences.

I think the work is very well organised, it would help to understand the mechanism of action of this novel antimicrobial agent but at this point some additional experiments should be carry out.

Reviewer #2 (Remarks to the Author):

The aim of this study is to investigate the mechanism of action of lugdunin on lipid membranes using both in vitro and in silico approaches. The affinity of lugdunin for different membranes was determined using the tryptophan blue shift of the maximum emission wavelength as a function of vesicle/lipid concentration. The obtained partition coefficients remained the same for POPC membranes with different amounts of PG lipids, while the 20% of cholesterol decreased its affinity by almost an order of magnitude. Interestingly, PG lipids enhanced the proton transport activity of lugdunin, whereas cholesterol hindered it. Spectroscopic experiments indicated the formation of parallel beta sheets on the DMPC membrane, which was further supported by experiments with methylated analogues. Finally, MD simulations showed transmembrane nanotubes composed of lugdunin. This is a good, detailed study with limited added value to general readership of Nature Communication. Moreover, there are several issues.

The membranes used do not mimic gram-positive bacteria or *Staphylococcus aureus*. Depending on the strain, *S. aureus* contains 10-40% PG lipids and 17-75% cardiolipin. There are no PC lipids that were used in the study. Furthermore, cardiolipin was completely omitted in the study. FTIR experiments are performed on DMPC membrane, which is inconsistent with the rest of the study.

It is unclear how the employed concentrations of the peptides and lipids correspond to the concentrations used in the previous study determining the toxicity and antibacterial activity. Therefore, the relevance of the partition coefficients is unclear. If all peptides are bound to membranes in the bacterial and toxicity experiments, partition coefficients at different concentrations may be irrelevant. Moreover, FTIR experiments suggested aggregation of lugdunin depending on its concentration. Therefore, it remains unclear whether the activity/toxicity is caused by different affinity to bacterial/eucaryotic membranes or if it is caused by different propensity to form nanopores on these membranes.

The lugdunin-membrane affinity and different propensity to form pore structures could be determined from MD simulations including the effects on membrane composition, but this has not been done. Only different preformed pores were studied, so the correct structure remains unclear including the propensity to form such structures. From S20 it is elusive if the build pore structures are unstable in the presence of cholesterol and bacterial lipids were not even tested.

The formation of conductive nanopores should be further supported by conductance measurements on black lipid membranes using different size of conductance species which would demonstrate the existence of pores, their lifetime, and the size.

We have addressed all issues raised by the reviewers point-by-point and have marked major changes and additional information in the main manuscript.

Reviewers' comments:

Reviewer #1 (Remarks to the Author):

The manuscript describe the work carried out to understand the mechanism of action of antimicrobial peptide lugdunum. This is a natural cyclic peptide with antimicrobial properties that was discovered few years ago (Nature 2016m 535, 511). The lack of new antimicrobial drugs is creating a main concern to health authorities. There are a number of antibiotic-resistant bacterial strains whose treatment is difficult. Therefore the development of new antimicrobial with different mechanism of action are required. At this respect this cyclic peptide represent a very promising candidate. Since the first publication a number of works have already study the mechanism of action of this AMP, finding the proton translocation as one of the main consequences of dissipation of the membrane potential. The authors try to evaluate this mechanism using artificial models. They propose a channel-type structure as responsible of the proton transfer based on the computational studies. Other studies, such as FTIR support the antiparallel-like structure of the cyclic peptide.

Regarding the computational studies, how many hydrogen ions support the proposed tubular structure? Although FTIR suggest the antiparallel stacking, did the author check the stability of this beta-sheet.

Thank you for your inquiry about the hydrogen bonds supporting our proposed tubular structure. This detail can be found in the Supplementary Information, Fig. 24, which illustrates the number and positions of intermolecular hydrogen bonds essential for the stability of the tubular structure, shown for preformed stacks with 3-5 lugdunin molecules, in two different configurations (see Supplementary Information, Fig. 23). We apologize for any oversight in directing attention to this figure in the main manuscript text.

Since the tubular structures display substantial dynamics, we counted the number of acceptor-donor pairs between the respective lugdunin molecules. Pairs are defined by acceptor-donor distances below 7 Å and acceptor-donor-hydrogen angles below 30°. In configurations that exhibited stability over microseconds (Supplementary Fig. S24), the number of pairs varied, generally ranging from 10 to 20. The observed fluctuations reflect the inherent dynamics within the lugdunin stacks.

Importantly, the antiparallel stacking was not only used as a starting configuration but was also observed for *spontaneously* formed lugdunin channels (simulation results are added in the revision, see also Supplementary Movie 1), in excellent agreement with our FTIR analysis.

How many water molecules are filling the channel of the tubular structure?

Thank you for your question regarding the number of water molecules within the channel of the tubular structure. This information is detailed in the main manuscript (Fig. 6, Section *Structure, orientation, and properties of lugdunin nanotubes in silico*). The figure illustrates the tubular structure of the lugdunin stack (Fig. 6a), and provides a quantitative analysis of the water dipole moment within the channel (Fig. 6b, c), the water density (Fig. 6d) and the number of water molecules within the channel (Fig. 6e, on average ~10 molecules). We apologize if this was not sufficiently emphasized in the text.

Why the channel is made only by four cyclic peptides? did the author evaluated five or six subunits? in fact, Ghadiri work with D,L-cyclic peptides already reported that the most commonly observed species are the five stacked subunits (Acc. Chem. Res 2013, 46, 2955).

We appreciate your reference to the significant work by Ghadiri et al., highlighting the prevalence of pentameric structures in D,L-cyclic peptides (even numbered cyclic peptide *cyclo*-[D-Leu-L-Gln-(D-Leu-L-Trp)₃]) as the most commonly observed species in L- α -lecithin bilayers at 1 M KCl. However, as

lugdunin shows distinct structural differences from the peptides introduced by Ghadiri et al., we cannot assume that five stacked subunits are also the most stable structure for lugdunin nanotubes.

Our all-atom molecular dynamics simulations revealed nuanced stability preferences for lugdunin stacks in various lipid bilayers: Analyses on preformed lugdunin stacks of 3-5 units (in two different configurations) within DMPC, DOPC, POPC, and POPC/Chol membranes revealed optimal stability with trimers in DMPC (310 K), tetramers in DOPC and POPC, and pentamers in POPC/Chol (298 K), detailed in the manuscript (Section *Structure, orientation, and properties of lugdunin nanotubes in silico*, see also Supplementary Information Fig. 23 and 24).

Inspired by the reviewer's comment, we now added all-atom MD simulations of spontaneous stack formation. Stacks that allowed the formation of closed water wires across the hydrophobic core of the membrane were built of 3 or 4 molecules. These findings, alongside simulation movies of spontaneous stack formation and the water dynamics within the stack, are now incorporated into the manuscript and the Supplementary Information, offering deeper insights into the peptide assembly dynamics under varying conditions.

To carry the studies in transport they have used N-methylated peptides following Ghadiri precedent, I would recommend to cite one of his work as pretenders (ref 42 in the manuscript that is only related with nanotube forming cyclic peptide (Nature 1994, 366, 324)).

We thank the reviewer for this suggestion and added the reference at the appropriate position. Of note, N-methylated lugdunin peptides do not prominently show the characteristic amide I_⊥ ($\approx 1630\text{ cm}^{-1}$) and amide II_⊥ ($\approx 1525\text{ cm}^{-1}$) bands suggesting that the N-methylated lugdunin derivatives do not form stable dimers in contrast to the even-numbered planar cyclopeptides of the Ghadiri group. This observation can be attributed to the non-planarity of the ring structure of lugdunin owing to the odd number of amino acids and the thiazolidine moiety, which renders the ring system more dynamic (see Supplementary Fig. S24).

I would also recommend to mention with more detail the synthetic procedure used to prepare this peptide. Also add the reference about the N-methylation on the resin. This is not a so general method. In the supporting information experimental data for the preparation of the peptides should be provided.

As pointed out by the reviewer, the N-methylation is not a trivial procedure and we added one reference regarding the synthesis of N-methylated cyclic peptides as suggested by the reviewer (reference [54] Chatterjee, J., Laufer, B. & Kessler, H. *Nat. Protoc.* **7**, 432–444 (2012)). The detailed experimental procedure is provided in the Supplementary Information accompanied by experimental data (chromatograms, Supplementary Figs. 1-3), fragment ion analysis, and ¹H- as well as ¹³C-NMR spectra (Supplementary Figs. 4-9).

Although the lugdunin has a similar structure to Ghadiri D,L- cyclic peptides or also to Manno-peptimycins (that is also a natural alternating D,L-cyclic peptide). the transport mechanism can be different.

There are several transport mechanism scenarios conceivable for cyclic peptides that are apolar in nature (manno-peptidimycins mentioned by the reviewer do not belong to this class, as these cyclic peptides are glycopeptide antibiotics) and thus can be inserted into the hydrophobic core of a lipid membrane: i) the cyclic peptide disturbs the membrane integrity and thus permeabilizes the membrane for polar or charged molecules/ions; ii) the cyclic peptide acts as a carrier, which diffuses across the membrane to transport polar or charged molecules/ions; iii) the cyclic peptide assembles in a structure that spans the membrane to form a channel through which polar or charged molecules/ions can pass.

From previous results (reference [12], Schilling, N. A. *et al. Angew. Chem. Int. Ed.* **58**, 9234–9238 (2019)), i) can be ruled out for lugdunin, as the insertion of lugdunin into membranes (bacterial and artificial ones) does not permeabilize the membrane for larger polar molecules.

ii) or iii) are both conceivable. It was the aim of this study to unravel this unknown structure of lugdunin in lipid membranes that can explain the observation of a lugdunin-mediated proton transfer across membranes. Our structural and functional analysis, and all-atom molecular dynamics simulations, revealed an assembly of peptides in the membrane forming a channel-like structure. Lugdunin derivatives not capable of forming the channel-like structure do not show proton transport.

The significance of the thiazolidine moiety for channel formation was addressed in additional free energy simulations of a lugdunin mutant, where the thiazolidine was replaced by alanine (lugdunin^{Cys4Ala}). This molecular alteration led to a substantial shift in the potential of mean force for peptide insertion into POPC membranes (see new Fig. 1e, f, Supplementary Figs. 19, 20). The partition coefficient is lowered by almost two orders of magnitude. Most strikingly, the minimum of ΔG for transfer of lugdunin^{Cys4Ala} into the membrane is shifted by 2-5 Å towards the membrane interfacial region, with ΔG additionally displaying a steeper increase towards the membrane core region (at $d = 0$). These changes in the free energy profile collectively act as a formidable barrier to the assembly of lugdunin stacks traversing the membrane.

Did the author checked if they are able to transport chloride? I would recommend to carry lucigenin transport experiments (halide sensor dye) to discharge that the peptides are not transporting HCl through a different mechanism but with similar consequences.

We appreciate the suggestion and performed the proposed lucigenin-based chloride transport assay. The results are shown in Supplementary Fig. 22 clearly demonstrating that lugdunin is not capable of transporting chloride.

Supplementary Fig. 22 | Fluorescence time course of lucigenin entrapped in lipid vesicles exposed to a chloride gradient of 10 mM in the absence or the presence of lugdunin (peptide-to-lipid ratio 1:250, n/n).

I think the work is very well organised, it would help to understand the mechanism of action of this novel antimicrobial agent but at this point some additional experiments should be carry out.

We thank the reviewer for their encouraging comments. We think that we have addressed all concerns raised by the reviewer by adding the requested additional experiments and simulations.

Reviewer #2 (Remarks to the Author):

The aim of this study is to investigate the mechanism of action of lugdunin on lipid membranes using both in vitro and in silico approaches. The affinity of lugdunin for different membranes was determined using the tryptophan blue shift of the maximum emission wavelength as a function of vesicle/lipid concentration. The obtained partition coefficients remained the same for POPC membranes with different amounts of PG lipids, while the 20% of cholesterol decreased its affinity by almost an order of magnitude. Interestingly, PG lipids enhanced the proton transport activity of lugdunin, whereas cholesterol hindered it. Spectroscopic experiments indicated the formation of parallel beta sheets on the DMPC membrane, which was further supported by experiments with methylated analogues. Finally, MD simulations showed transmembrane nanotubes composed of lugdunin. This is a good,

detailed study with limited added value to general readership of Nature Communication. Moreover, there are several issues.

We thank the reviewer for their critical evaluation. However, we disagree with the statement that the study is of limited added value. Lugdunin is the first drug-like molecule found in bacteria of the human nose and has been the founding member of a completely new class of cyclopeptides, called fibupeptides, with an odd number of amino acids and its surprising thiazolidine moiety within the macrocycle – a unique feature, shown to be invaluable for its antimicrobial activity. Which new mechanism nature invented for lugdunin’s function in the human nose and which structure lugdunin forms in membranes is as yet unknown. With our study, we are the first who show that lugdunin forms a stacked membrane channel with an aligned water channel that allows the passage of protons and monovalent cations but no anions. This finding is a breakthrough concerning our understanding of this new class of fibupeptides. It is the first report on a natural compound exerting ion translocation via a stacked channel-like structure which was up to now only shown for purely synthetic peptides with a different architecture (fully planar ring geometry with an even number of D,L-amino acids and no thiazolidine moiety). Understanding this mechanism will lead to new design principles of antimicrobial peptides and ultimately medical function.

The membranes used do not mimic gram-positive bacteria or *Staphylococcus aureus*. Depending on the strain, *S. aureus* contains 10-40% PG lipids and 17-75% cardiolipin. There are no PC lipids that were used in the study. Furthermore, cardiolipin was completely omitted in the study. FTIR experiments are performed on DMPC membrane, which is inconsistent with the rest of the study.

We thank the reviewer for this critical discussion of our employed lipid compositions and would like to answer this comment stepwise:

Use of PC lipids: Generally, POPC is mostly used in *in vitro* (vesicle) assays as it provides a fluid and non-charged lipid species. It has been used in our initial study that provided the first evidence that lugdunin is capable of transporting protons (reference [12], Schilling, N. A. *et al. Angew. Chem. Int. Ed.* **58**, 9234–9238 (2019)). Here, we used this lipid as a starting point to relate the previously observed proton translocation with the partitioning of lugdunin in POPC lipid bilayers. We then extended the lipid compositions according to the main components of Gram-positive bacterial membranes (e.g., reference [24] Epanand, R. F., Savage, P. B. & Epanand, R. M. *Biochim. Biophys. Acta Biomembr.* **1768**, 2500–2509 (2007)) and eukaryotic membranes (e.g., reference [25] van Meer, G., Voelker, D. R. & Feigenson, G. W. *Nat. Rev. Mol. Cell Biol.* **9**, 112–124 (2008)).

Addition of cardiolipin: Besides PG, we have now added cardiolipin as the second major component of Gram-positive bacteria in our study as suggested by the reviewer. We either added POPG or cardiolipin to a POPC bilayer to investigate their influence independently. For both lipid components, the partition coefficients remain in the same regime ($K_{\chi} \gg 10^5$, Tab. 1). Also the proton translocation activity of lugdunin in POPG/POPC or cardiolipin/POPC membranes remains the same (Fig. 5a).

Fig. 5 | Proton transport activity of lugdunin and its methylated analogues 3-5 measured with pyranine entrapped in lipid vesicles. a Proton translocation induced by lugdunin as a function of the lipid composition. A nominal peptide-to-lipid ratio of 1:250 (n/n) was applied in all cases.

Use of DMPC for ATR-FTIR experiments: 1,2-Dimyristoyl-*sn*-glycero-phospholipids are standard lipids in structural and orientational IR studies (e.g., references [33] Claro, B. *et al. Colloids and surfaces. B, Biointerfaces* **196**, 111349 (2020) and [41] Claro, B. *et al. Colloids and surfaces. B, Biointerfaces* **208**, 112086 (2021)). While the use of longer and unsaturated fatty acids like POPC would be more physiological, POPC lipids do not form sufficiently ordered multi-bilayers that allow for a reliable determination of the angle between the lipids and the proposed axis of the channel-like structure formed by lugdunin. This angle was, however, used as further support for the formed nanotubes in lipid bilayers and was corroborated by MD simulations. To rationalize this aspect and provide data, we added Supplementary Table 8. The provided orientational parameters of different multi-bilayers show that POPC-multi-bilayers are much less ordered than DMPC and DPPC multi-bilayers. A peptide nanotube in POPC multi-bilayers is expected to show a tilt angle of around $\sim 55^\circ$, which is very close to the magic angle found for a random orientation of the peptide rings. We have chosen DMPC and not DPPC, as DMPC is still in the fluid phase (as POPC) and not in the gel phase as DPPC at room temperature.

Supplementary Table 8 | Dichroic ratio, order parameter, and effective tilt angle of pure lipid multi-bilayers composed of POPC, DPPC, or DMPC ($n \geq 2$ experiments, mean \pm standard deviation).

Lipid	R^{ATR}	$S(\theta)$	$\theta / ^\circ$
POPC	1.56 ± 0.02	0.29 ± 0.02	43.4 ± 0.5
DPPC	1.12 ± 0.02	0.68 ± 0.02	27.4 ± 0.6
DMPC	1.12 ± 0.01	0.69 ± 0.01	27.2 ± 0.5

It is unclear how the employed concentrations of the peptides and lipids correspond to the concentrations used in the previous study determining the toxicity and antibacterial activity. Therefore, the relevance of the partition coefficients is unclear. If all peptides are bound to membranes in the bacterial and toxicity experiments, partition coefficients at different concentrations may be irrelevant.

The partition coefficients indicate that lugdunin strongly partitions into membranes. The reviewer is right that a partition coefficient on the order of 10^5 means that more than 99.999 % of the added peptide concentration is located in the membrane. This partition coefficient correlates to the previous studies in the way that all added lugdunin molecules will almost exclusively be located in the bacterial membranes – independent of the employed nominal concentration demonstrating how strongly lugdunin penetrates into lipid membranes. However, partitioning is only one parameter governing lugdunin activity (see also below). To form peptide nanotubes, lugdunin also has to reach the core of a lipid membrane which we strengthened by the potential of mean force simulations (new Fig. 1e, f, Supplementary Figs. 19, 20). How strongly membrane properties affect lugdunin partitioning and the ability to insert deeply into the membrane was shown by *in silico* and *in vitro* measurements of membranes harboring 20 mol% of the eukaryotic-specific component cholesterol. Increasing the cholesterol content from 0 to 20 mol% reduces the partition coefficient and creates an increased energetical barrier for lugdunin insertion indicating that cholesterol influences the peptide interaction with a membrane and thus is a relevant parameter for our understanding of the peptide's toxicity.

Moreover, FTIR experiments suggested aggregation of lugdunin depending on its concentration. Therefore, it remains unclear whether the activity/toxicity is caused by different affinity to bacterial/eukaryotic membranes or if it is caused by different propensity to form nanopores on these membranes.

Indeed, we suggest that the activity of lugdunin towards bacterial cells is dictated by both the affinity of the peptide for different cell membranes and its propensity to form peptide nanopores leading to a proposed two-step mechanism. To illustrate this, we added the new Fig. 8 to the main text. The insertion into a lipid membrane (addressed via tryptophan fluorescence experiments) is followed by the self-assembly of lugdunin to peptide nanotubes (derived from the IR-spectroscopic measurements and molecular dynamics simulations) spanning the membrane and acting as membrane channels (obtained from voltage-clamp experiments on BLMs). The toxicity of lugdunin is consequently governed by several parameters which could influence one of the mentioned processes (like *N*-methylation impeding the ability to form nanotubes) or both of them (like a lipid condensing effect hindering peptide insertion and assembly).

Fig. 8. Proposed two-step mechanism of lugdunin activity. To exert its mechanism of action, lugdunin first inserts into lipid membranes and then self-assembles into peptide nanotubes via intermolecular hydrogen bonds. Both processes are vital for lugdunin's antibacterial activity and are affected by different membrane and peptide properties.

As our results are the very first that show that lugdunin forms peptide nanotubes, our findings do indeed raise new questions about membrane selectivity, which will be addressed in the future.

The observed aggregation of lugdunin in the interbilayer space visible in the IR-spectroscopic data (given by the high peptide-to-lipid ratio to obtain a good signal-to-noise ratio), does not influence the peptide activity, as 3,6-dityryptophan-lugdunin does not show these aggregated structures but even displays an enhanced proton translocation ability.

The lugdunin-membrane affinity and different propensity to form pore structures could be determined from MD simulations including the effects on membrane composition, but this has not been done. Only different preformed pores were studied, so the correct structure remains unclear including the propensity to form such structures. From S20 it is elusive if the build pore structures are unstable in the presence of cholesterol and bacterial lipids were not even tested.

We express our gratitude to the reviewer for the constructive criticism. In our revised manuscript, we have thoroughly addressed both identified issues: (1) the quantification of lugdunin's affinity for various membranes, and (2) the unbiased assembly of lugdunin into tubular structures.

- (1) Concerning the affinity of lugdunin for different membrane compositions, we conducted additional all-atom umbrella simulations that enabled the determination of the potential of mean force (PMF). Our analysis encompassed four distinct systems: lugdunin's insertion into i) POPC, ii) into POPC/Chol membranes, iii) into a representative model of a Gram-positive bacterial membrane and iv) of a lugdunin analogue into POPC, where the thiazolidine moiety is substituted with alanine.

The partition coefficient derived from the PMF (Fig. 1e, f, Tab. 1, Supplementary Figs. 19, 20) of $\log_{10}(K\chi) = 5.6$ for pure POPC membranes was decreased by one order of magnitude in the presence of cholesterol to $\log_{10}(K\chi) = 4.6$, in excellent agreement to our Trp-fluorescence *in vitro* experiments. Noteworthy, the replacement of the thiazolidine moiety by alanine resulted in a reduction of the partition coefficient by almost two orders of magnitude ($\log_{10}(K\chi) = 3.7$). Such a substantial shift underscores the pivotal role of the thiazolidine moiety in the interaction dynamics between lugdunin and lipid membranes. We have incorporated these pivotal findings into the results section of our manuscript.

- (2) Additionally, we now report the unbiased simulations for the spontaneous insertion of lugdunin into the membranes *and* their spontaneous assembly to tubular structures spanning

the hydrophobic core of the membrane (Supplementary Movie 1, 2, and a paragraph added to the conclusions section).

Fig. 1 | Insertion of lugdunin into the membrane phase. e-f Potential of mean force profiles for the insertion of lugdunin (**1**) and the analogue lugdunin^{Cys4Ala} (Supplementary Fig. 19a) into POPC (e) and POPC/cholesterol (80/20, *n/n*) model membranes (f). Shaded areas represent 95% confidence intervals derived from bootstrapping of umbrella simulation histograms.

The formation of conductive nanopores should be further supported by conductance measurements on black lipid membranes using different size of conductance species which would demonstrate the existence of pores, their lifetime, and the size.

As suggested by the reviewer, we performed voltage-clamp experiments on BLMs to record single-channel activity. Since the concentration of protons is naturally extremely low (10^{-7} M) under physiological conditions, proton transport cannot be measured on the single-channel level. As the proton transport in the vesicle assay does not lead to the establishment of a membrane potential that would halt the H^+ -transport, and chloride is not transported as a counter ion, we hypothesized that lugdunin nanopores transport other monovalent cations. We thus recorded transmembrane current traces in the presence of 500 mM KCl, NaCl, or CsCl (new Fig. 7 and Supplementary Fig. S26). We found single-channel events in the presence of lugdunin. The conductance values follow the order of $Na^+ < K^+ < Cs^+$ with dwell time constants between 470-700 ms. This is in agreement with the expected behavior for nanopores as their ion selectivity is mainly governed by the hydrodynamic radius of the respective ion. Furthermore, the obtained conductance values allowed a comparison with the pore-forming peptide gramicidin A demonstrating that the lugdunin nanotube would have an effective diameter of around 4 Å, which is in line with our results obtained from simulations. We added this information to the manuscript.

Fig. 7 | Single-channel properties of membrane channels formed by lugdunin nanotubes. **a** Recorded transmembrane currents across BLMs composed of POPC reveal single-channel conductance events in the presence of 500 mM Na⁺, K⁺, and Cs⁺ at a voltage of +100 mV. **b** Event histogram ($n = 704$ events, bin width = 1.15 pS) of the K⁺-conductance. Three Gaussian functions were fitted to the histogram revealing maximum conductance values of 19.9 pS, 24.3 pS and 29.3 pS. **c** Dwell time histogram of single-channel events in the presence of K⁺ ($n = 701$ events, bin width = 150 ms, dwell times larger than 5000 ms were excluded). A mono-exponential fit yields a dwell time constant of (479 ± 15) ms.

We thank the reviewer for their consideration of our work. We think that with the more elaborated explanation of lugdunin's insertion and assembly in membranes and with the supporting voltage-clamp experiments we have addressed all concerns raised by the reviewer.

Reviewer #1 (Remarks to the Author):

The authors have addressed most of the doubts raised during the previous review. The ion transport and spectroscopic characterization provided by the authors support the interpretation and conclusions of their studies. However, the least convincing data comes from computational studies. They mention that the tubular structure is supported by a number of acceptor-donor interactions that varies between 10 to 20 pairs but considering a distance less than 7 Å (mentioned in the response letter). This is a very long distance for a moderately stable hydrogen bond; these distances are usually around 3 Å and, in any case, they are shorter than 3.5 Å. Therefore, the number of these interactions must be smaller. How can this tubular structure be supported by a small number of hydrogen bonding interactions? The authors must clarify this point and include the number of these interactions, considering the typical distances for this type of non-covalent bonds, in the manuscript.

The second doubt refers to the transport of cations. The BLM clearly supports such transport and the preference for Cs ions over smaller alkaline cations. But looking at the water channel illustrated in the movie attached as supporting data (435079_1_video_8269011_s3k24g), there is a very narrow channel, in which the water molecules are forming an allied row along the channel of the tubular structure. In previous computational studies with nanotubes by Ghadiri and others, cation transport is carried out with hydrated ions, consequently requiring a larger channel diameter. Have the authors carried out any computational study including ions? Are the ions able to go into the nanotube channel? Can the authors provide a convincing answer to this discrepancy?

Once these aspects are clarified, the article can be published

Reviewer #2 (Remarks to the Author):

The study reveals the structure explaining the mechanism of action lugdunin a new cyclopeptide with antimicrobial properties. An important finding in the field of antimicrobial cyclic peptides. The authors performed a number of additional experiments and simulations. This is a really thorough study with a lot of work and nice data in it. However, the main issue of the manuscript was not fully addressed.

1) "The membranes used do not mimic gram-positive bacteria or *Staphylococcus aureus*. Depending on the strain, *S. aureus* contains 10-40% PG lipids and 17-75% cardiolipin. There are no PC lipids that were used in the study. Furthermore, cardiolipin was completely omitted in the study. FTIR experiments are performed on DMPC membrane, which is inconsistent with the rest of the study."

New experiments added mixture of cardiolipin, specifically POPC:cardiolipin (1:1) mixture, but this is not bacterial mimic. The main point remains not using a bacterial mimic and using POPC lipids, which are not present in significant amounts in gram-positive bacteria or *Staphylococcus aureus*. Therefore, this issue has not been resolved in my view and the authors did not explain why they are not using lipid mixtures relevant to explain the antibacterial activity.

There is a new Figure 8 illustrating a possible mechanism. I would like to draw the authors' attention to this, because lugdunin seems to be inserted from both sides of the membranes in the first step. However, when it is introduced into bacteria, it is only present on the outside of the bacteria. Therefore, the aggregation/assembly on the membrane is probably asymmetric. The image provided seems more likely to refer to some other *in vitro* experiment with lugdunin introduced on both sides of the membrane.

Reviewers' comments:

Reviewer #1 (Remarks to the Author):

The authors have addressed most of the doubts raised during the previous review. The ion transport and spectroscopic characterization provided by the authors support the interpretation and conclusions of their studies. However, the least convincing data comes from computational studies. They mention that the tubular structure is supported by a number of acceptor-donor interactions that varies between 10 to 20 pairs but considering a distance less than 7 Å (mentioned in the response letter). This is a very long distance for a moderately stable hydrogen bond; these distances are usually around 3 Å and, in any case, they are shorter than 3.5 Å. Therefore, the number of these interactions must be smaller. How can this tubular structure be supported by a small number of hydrogen bonding interactions? The authors must clarify this point and include the number of these interactions, considering the typical distances for this type of non-covalent bonds, in the manuscript,

We appreciate the reviewer's comments and the opportunity to clarify this aspect of our study. In our original response, **Supplementary Fig. 24** depicted donor-acceptor pairs within a 7 Å range. This range was chosen to include two distinct shells of polar interactions, as illustrated in **Fig. R1**. The first shell, below 4 Å, represents typical hydrogen bond distances, with the maximum distance observed at 2.94 Å (as compared to 2.88 Å determined from ATR-FTIR spectroscopy). The second shell, spanning 4-7 Å, encompasses broader polar interactions extending to the next donor or acceptor within the cyclic peptide structure.

Fig. R1 | Distance distribution of attractive polar interactions between backbone atoms of two peptides in stable trimeric channels.

To address the reviewer's concern, we have revised **Supplementary Fig. 24** to focus solely on donor-acceptor pairs within the hydrogen bonding range of 4 Å or less. This revision reveals that the number of hydrogen bonds stabilizing two stacked lugdunin molecules typically ranges between 2 to 6 (compare representative snapshot for a trimeric channel below). While this may appear to be a modest number, our data suggests that these hydrogen bonds are surprisingly effective in stabilizing (meta)stable tubular structures across the hydrophobic core of the membrane. This observation is further supported by the spontaneous formation of lugdunin channels we recorded.

We propose that the strength of these hydrogen bonds is possibly enhanced by the low dielectric constant environment within the membrane's hydrophobic core, approximately 2, as suggested by previous studies (e.g., Böckmann et al., *Biophys. J.* **2008**, 95, 1837-1850). This enhancement in bond strength within such an environment might compensate for the lower number of interactions.

For context, similar phenomena are observed in other systems. For example, gramicidin A dimer structures (see also below) are known to be stabilized by in total just 2-6 hydrogen bonds (Sun et al., *JCTC* **2020**, 17, 7-12). This comparison highlights that a small number of hydrogen bonds can be sufficient for structural stability, particularly in specific microenvironments like the hydrophobic core of a membrane.

We hope this clarification better explains our findings and strengthens the interpretation of our computational studies. We have incorporated these details into the manuscript to provide a more comprehensive understanding of the structural dynamics at play.

Fig. R2 | Comparative snapshots of the hydrogen bond network in a trimeric lugdunin stack in a DMPC bilayer after **a** 0 μ s, **b** 1.42 μ s, and **c** 2.77 μ s.

The second doubt refers to the transport of cations. The BLM clearly supports such transport and the preference for Cs ions over smaller alkaline cations. But looking at the water channel illustrated in the movie attached as supporting data (435079_1_video_8269011_s3k24g), there is a very narrow channel, in which the water molecules are forming an allied row along the channel of the tubular structure. In previous computational studies with nanotubes by Ghadiri and others, cation transport is carried out with hydrated ions, consequently requiring a larger channel diameter. Have the authors carried out any computational study including ions? Are the ions able to go into the nanotube channel? Can the authors provide a convincing answer to this discrepancy?

We thank the reviewer for their remark and would like to point out that, while lugdunin's structure resembles the one of synthetic D,L-cyclopeptides, its pore geometry is more related to the well-studied membrane channel gramicidin A from *Bacillus brevis*. Gramicidin A forms a 4 Å-wide pore in lipid membranes (Kelkar et al., *Biochim. Biophys. Acta Biomembr.* **2007**, 1768, 2011-2025) accommodating a single file of water molecules (Finkelstein et al., *J. Membr. Biol.* **1981**, 59, 155-171; Urban et al., *Biochim. Biophys. Acta Biomembr.* **1980**, 602, 331-354) and allowing the passage of metal cations in the affinity order of $\text{Cs}^+ > \text{Na}^+ > \text{K}^+$ (Hladky et al., *Biochim. Biophys. Acta Biomembr.* **1972**, 274, 294-312; Neher et al., *J. Membr. Biol.* **1978**, 40, 97-116) - just like lugdunin. The translocation of ions through this pore is realized by partial dehydration of the ions at the channel mouth (Poxleitner et al., *Z. Naturforsch.* **1993**, 48, 654-665). Consequently, inside the channel, the ions are only hydrated by two water molecules and are re-solvated by the peptide's carbonyl groups. For lugdunin, we found a similar ion conductance suggesting that it also only hosts a single column of water molecules which is in line with the observations from our MD simulations. Similar to gramicidin A, we suspect that lugdunin also translocates metal ions in a partially dehydrated state by stabilizing them via its carbonyl groups. In this state, the narrow channel tube is sufficient to mediate ion translocation. In contrast, the synthetic D,L-cyclopeptides introduced by Ghadiri and co-workers result in considerably larger pores (> 7.5 Å depending on the exact ring size; Ghadiri et al., *Nature* **1994**, 369, 301-304) and have

therefore shown to accommodate more than one water column (Engels et al., *J. Am. Chem. Soc.* **1995**, *117*, 9151-9158). The larger pore size also results in a larger ion conductance obtained from voltage-clamp experiments (Ghadiri et al., *Nature* **1994**, *369*, 301-304). In this scenario, there is no need for an extensive dehydration of the metal ions to be translocated through the channel. Conclusively, the discrepancy described by the reviewer can be resolved by considering a similar transport mechanism as described for gramicidin A. We illustrate the differences between the three peptides regarding their size, ion conductance, and water diffusion in **Fig. R3**.

Fig. R3 | Comparison of the pore size, ion conductance, and water diffusion of peptide pores constituted of gramicidin A, lugdunin, and synthetic D,L-cyclopeptides. Small channel diameters (gramicidin and lugdunin) result in partial ion dehydration, while larger diameters can accommodate ions in their hydrated form (synthetic cyclopeptide).

Once these aspects are clarified, the article can be published

We thank the reviewer for their considerate questions and hope that we could clarify the remaining points.

Reviewer #2 (Remarks to the Author):

The study reveals the structure explaining the mechanism of action lugdunin a new cyclopeptide with antimicrobial properties. An important finding in the field of antimicrobial cyclic peptides. The authors performed a number of additional experiments and simulations. This is a really throw study with a lot of work and nice data in it. However, the main issue of the manuscript was not fully addressed.

1) "The membranes used do not mimic gram-positive bacteria or *Staphylococcus aureus*. Depending on the strain, *S. aureus* contains 10-40% PG lipids and 17-75% cardiolipin. There are no PC lipids that were used in the study. Furthermore, cardiolipin was completely omitted in the study. FTIR experiments are performed on DMPC membrane, which is inconsistent with the rest of the study." New experiments added mixture of cardiolipin, specifically POPC:cardiolipin (1:1) mixture, but this is not bacterial mimic. The main point remains not using a bacterial mimic and using POPC lipids, which are not present in significant amounts in gram-positive bacteria or *Staphylococcus aureus*. Therefore,

this issue has not been resolved in my view and the authors did not explain why they are not using lipid mixtures relevant to explain the antibacterial activity.

We thank the reviewer for their comment. Following the reviewer's suggestion, we now also determined the partition coefficient for bilayers composed of only POPG and cardiolipin. To fully mimic the cell membrane of *S. aureus*, we chose a lipid composition of POPG/cardiolipin (50:50, *n/n*), lacking POPC. We found a blue shift of $\Delta\lambda_{\max} = (19 \pm 3)$ nm and a partition coefficient $\log_{10}(K_X) = 5.7 \pm 0.2$ very similar to the other employed lipid compositions except for those containing cholesterol (Table 1). We also performed the proton translocation assay with POPG/cardiolipin (50:50, *n/n*) (Fig. 5a) demonstrating the fast dissipation of the proton gradient in the presence of lugdunin with the same kinetics as that found for POPC/POPG (50:50, *n/n*).

Fig. 5 | Proton transport activity of lugdunin and its methylated analogues 3-5 measured with pyranine entrapped in lipid vesicles. a Proton translocation induced by lugdunin as a function of the lipid composition. **b** Proton translocation induced by 3,6-ditryptophan-lugdunin and methylated analogues 3, 4, and 5. A nominal peptide-to-lipid ratio of 1:250 (*n/n*) was applied in all cases.

There is a new Figure 8 illustrating a possible mechanism. I would like to draw the authors' attention to this, because lugdunin seems to be inserted from both sides of the membranes in the first step. However, when it is introduced into bacteria, it is only present on the outside of the bacteria. Therefore, the aggregation/assembly on the membrane is probably asymmetric. The image provided seems more likely to refer to some other in vitro experiment with lugdunin introduced on both sides of the membrane.

We revisited the figure and changed the position of the individual peptide rings to illustrate that they were initially added from one side of the membrane (Fig. 8).

Fig. 8. Proposed two-step mechanism of lugdunin activity. To exert its mechanism of action, lugdunin first inserts into lipid membranes and then self-assembles into peptide nanotubes via intermolecular hydrogen bonds. Both processes are vital for lugdunin's antibacterial activity and are affected by different membrane and peptide properties.

Reviewer #1 (Remarks to the Author):

The author have carried out additional experiments to clarify some of the questions proposed. The number of hydrogen bonds that holds the tubular structure drops significantly. It is true that the strength of hydrogen bonds increases in the lipid medium, but it is also true that the pressures of the membranes, especially between their two layers, are so important that they can break the interactions between elements located in different layers.

In the case of gramicidin (GA), there are only two subunits that must be held together, so it would be less affected than the channel here reported. However, the opening times of the channels illustrated in Figure 7 are quite long, I think even longer than those observed for GA. The authors should explain this behaviour considering the reduced number of hydrogen bonds that are formed between the subunits.

With respect to transport, although the diameter of GA and cyclic peptides could be similar, the GA channel, being helical, can increase to adjust to the passage of ions, while those presented in the work are determined by the size of the macrocycle and that is fixed. How do you explain that Cs is transported 5 times faster than Na? This is something not very common in this type of channels in which the transport of K and Cs is usually quite similar. On the other hand, looking at the event histogram that appears in figure 7, we see three maxima at 19.9, 24.3 and 29.3 for the transport of K, the authors can explain this behaviour.

In general, the authors have addressed all aspects proposed by the reviewers, although the aspects of ion transport, as opposed to possible proton transport or other mechanism, must be clarified before it can be accepted.

Reviewer #2 (Remarks to the Author):

All my comments/issues were properly addressed and manuscript modified accordingly.

REVIEWERS' COMMENTS

Reviewer #1 (Remarks to the Author):

The author have carried out additional experiments to clarify some of the questions proposed. The number of hydrogen bonds that holds the tubular structure drops significantly. It is true that the strength of hydrogen bonds increases in the lipid medium, but it is also true that the pressures of the membranes, especially between their two layers, are so important that they can break the interactions between elements located in different layers.

We appreciate the reviewer's insightful comments and the opportunity to further clarify the stability of hydrogen bonds in the context of membrane pressures and their impact on the tubular structure reported in our study.

Indeed, as highlighted, the lateral pressure within membrane bilayers can be substantial (up to a few hundred bars), potentially influencing protein interactions and stability. However, we emphasize that, in equilibrium, the pressure normal to the membrane surface is constant and significantly lower (at 1 bar) than lateral membrane pressures (Vanegas *et al. J. Chem. Theor. Comput.* **2014**, *10*, 691-702). This distinction is critical for understanding the differential impact of membrane pressures on complex stability and interactions.

The high lateral membrane pressure may impede protein insertion within the membrane hydrophobic core region (high pressure, 'overpacked' region), but facilitate packing of transmembrane helices after insertion (see e.g. review by K. Corin and J. Bowie, *How physical forces drive the process of helical membrane protein folding*, *EMBO reports* **2020**, *23*, e53025).

For hydrogen bonds formed within the lateral membrane plane, such as those between transmembrane helices, the impact of high lateral pressures is indeed a concern. Conversely, the hydrogen bonds between lugdunin molecules stacked along the membrane normal (our study) are oriented perpendicular to the membrane plane. This orientation subjects them to the normal component of the membrane pressure, which, as mentioned, remains comparatively low and constant. Therefore, we argue that these bonds are less susceptible to the destabilizing effects of high lateral pressures, contributing to the stability of the tubular structure we observed.

In the case of gramicidin (GA), there are only two subunits that must be held together, so it would be less affected than the channel here reported. However, the opening times of the channels illustrated in Figure 7 are quite long, I think even longer than those observed for GA. The authors should explain this behaviour considering the reduced number of hydrogen bonds that are formed between the subunits.

We appreciate the reviewer's point regarding the need for an explanation of the observed differences in channel opening times between gramicidin A (GA) previously observed and the lugdunin-based channels described in our manuscript.

Gramicidin A and lugdunin-based channels indeed exhibit fundamentally different structural configurations and thus interactions with their lipid environments, leading to variations in their gating kinetics and energetics. GA forms a head-to-head helical dimer, primarily involving two subunits, while our lugdunin-based channels comprise 3-5 stacked cyclic peptides, introducing a more complex multimeric interaction within the membrane.

The specific lipid environment plays a crucial role in the formation and characteristics of these channels. For instance, studies such as those of Lundbaek *et al.* (*J. R. Soc. Interface* **2010**, *7*, 373-395) describe how gramicidin channels probe their lipid surroundings. Similarly, our research on the lugdunin-based channels, as detailed in Table 1, Figures 5, and Supplementary Figures 17, 18, 20, 24, 26, as well as Supplementary Table 8, indicates a significant dependence of the lugdunin stacks on the lipid environment, potentially leading to the observed variation in opening times when compared to GA. Given the intricate nature of these interactions, we have refrained from making a direct comparison of the energetics and kinetics of channel gating between these two types.

With respect to transport, although the diameter of GA and cyclic peptides could be similar, the GA channel, being helical, can increase to adjust to the passage of ions, while those presented in the work are determined by the size of the macrocycle and that is fixed. How do you explain that Cs is transported 5 times faster than Na? This is something not very common in this type of channels in which the transport of K and Cs is usually quite similar.

We thank the reviewer for the opportunity to further elaborate on our proposed mechanism of transport. The high conductance determined for Cs⁺ can indeed be explained accurately by the idea of a lugdunin nanotube and is also found for a conductive gramicidin dimer, both filled with a single file of water molecules. For gramicidin, it is well documented that the transport rate for different cations follows the lyotropic series (Cs⁺ > K⁺ > Na⁺ > Li⁺, e.g., Hladky *et al.*, *Biochim. Biophys. Acta Biomembr.* **1972**, *274*, 294-312; Myers *et al.*, *Biochim. Biophys. Acta Biomembr.* **1972**, *274*, 313-322; Neher *et al.*, *J. Membr. Biol.* **1978**, *40*, 97-116.). The lyotropic series is a result of the increasing dehydration enthalpies of the monovalent cations (Table R1).

Table R1. Dehydration enthalpies for the different ion species. Data were taken from House *et al.* (*Thermochim. Acta* **1983**, *66*, 365-368).

Ion species	$\Delta H_{\text{dehydration}} / \text{kJ mol}^{-1}$
Li ⁺	503
Na ⁺	410
K ⁺	329
Cs ⁺	264

Since the gramicidin dimer has a diameter of 4 Å (e.g., Kelkar *et al.*, *Biochim. Biophys. Acta Biomembr.* **2007**, *1768*, 2011-2025; Smart *et al.*, *Biophys. J.* **1993**, *65*, 2455-2460; Finkelstein *et al.*, *J. Membr. Biol.* **1981**, *59*, 155-171), which only decreases slightly upon potassium ion passage (Siu, Böckmann, *J. Phys. Chem. B* **2009**, *113*, 3195-3202), metal ions must be partially dehydrated to fit into the channel. This initial dehydration is decisive and limits the rate at which the ion can pass through the channel (Poxleitner *et al.*, *Z. Naturforsch. C, J. Biosci.* **1993**, *48*, 654-665). Once in the channel, the ions are partially re-solvated by the peptide's carbonyl groups and a water molecule located in the front and behind the ion (Fig. R1A).

The dehydration enthalpies restrict their translocation through a water-filled channel in general. Not only gramicidin but also a synthetic cyclopeptide introduced by Ghadiri and coworkers (Montenegro

et al., *Acc. Chem. Res.* 2013, **46**, 2955-2965) shows the same trend (Fig. R1). The same trend in normalized conductance is found for lugdunin.

Fig. R1. A. Orientation and hydration of a metal ion M^+ in the GA channel. The figure is based on the simulations performed by Poxleitner et al. (*Z. Naturforsch. C, J. Biosci.* **1993**, *48*, 654-665). **B.** Comparison of the normalized conductances of GA (Hladky et al., *Biochim. Biophys. Acta Biomembr.* **1972**, *274*, 294-312; Neher et al., *J. Membr. Biol.* **1978**, *40*, 97-116), the synthetic cyclopeptide cyclo[L-Trp-D-Leu]₄ (Montenegro et al., *Acc. Chem. Res.* 2013, **46**, 2955-2965) and lugdunin (mean conductances obtained from the respective event histograms).

On the other hand, looking at the event histogram that appears in figure 7, we see three maxima at 19.9, 24.3 and 29.3 for the transport of K⁺, the authors can explain this behaviour.

We appreciate the reviewer's inquiry about this interesting observation in our conductance data. In fact, we consciously decided not to focus on the three different conductance states in our manuscript, as it would extend the aim of the manuscript. However, the existence of the different conductance states can be explained by the presence of different lugdunin nanotube species differing in their respective number of constituting peptide monomers. These nanotube ensembles display a different length l which can be estimated by parameters obtained from MD simulations, allowing the calculation of the channel conductance G dependent on the channel length l by the Nernst-Planck electrodiffusion equation:

$$G = \left(\frac{z^2 F^2}{RT} \right) \left(\frac{(D_+ + D_-) c A}{l} \right).$$

Here, z is the ion charge, T is the temperature, D_+ and D_- are the cationic and anionic diffusion coefficients, c is the molar concentration, A is the channel cross-section area and F and R are denoted as usual. We determined the channel cross-section area from the peptide ring diameter ($2r = 3.66 \text{ \AA}$) obtained from MD simulations presented in the manuscript. The length l was calculated as $l = nd + 2d$, with n the number of peptide rings and $d = 0.56 \text{ nm}$ the distance between the peptide monomers. Using these parameters results in normalized conductance values for peptide nanotubes composed of 4 (1.00), 5 (0.80), and 6 (0.67) peptide monomers. These values are in good agreement with the distribution of the three Gaussian functions with channel conductances for K⁺ obtained from the event histogram displayed in Figure 7b (1.00, 0.83, and 0.68), thus supporting the model of dynamic nanotubular structures composed of different numbers of peptide monomers. We decided not to add this aspect to the manuscript, as the analysis is based on a rather simple approximation, and we would like to support this in the future with additional data. We hope that we could satisfy the reviewer with this deepened explanation.

In general, the authors have addressed all aspects proposed by the reviewers, although the aspects of ion transport, as opposed to possible proton transport or other mechanism, must be clarified before it can be accepted.

Reviewer #2 (Remarks to the Author):

All my comments/issues were properly addressed and manuscript modified accordingly.